# Distinct dynein complexes defined by DYNLRB1 and DYNLRB2 regulate mitotic and male meiotic spindle bipolarity

Shuwen He[1], John P. Gillies[2], Juliana L. Zang[2], Carmen M. Córdoba-Beldad[1], Io Yamamoto[1], Yasuhiro Fujiwara[3], Julie Grantham[1], Morgan E. DeSantis [2] & Hiroki Shibuya [1] ✉

Spindle formation in male meiosis relies on the canonical centrosome system, which is distinct from acentrosomal oocyte meiosis, but its specific regulatory mechanisms remain unknown. Herein, we report that DYNLRB2 (Dynein light chain roadblock-type-2) is a male meiosis-upregulated dynein light chain that is indispensable for spindle formation in meiosis I. In *Dynlrb2* KO mouse testes, meiosis progression is arrested in metaphase I due to the formation of multipolar spindles with fragmented pericentriolar material (PCM). DYNLRB2 inhibits PCM fragmentation through two distinct pathways; suppressing premature centriole disengagement and targeting NuMA (nuclear mitotic apparatus) to spindle poles. The ubiquitously expressed mitotic counterpart, DYNLRB1, has similar roles in mitotic cells and maintains spindle bipolarity by targeting NuMA and suppressing centriole overduplication. Our work demonstrates that two distinct dynein complexes containing DYNLRB1 or DYNLRB2 are separately used in mitotic and meiotic spindle formations, respectively, and that both have NuMA as a common target.

Cytoplasmic dynein 1 (dynein) is a microtubule (MT) motor protein complex that is responsible for retrograde transport of cellular cargos such as proteins, RNAs, and organelles[1,2]. Dynein is composed of four subunit classes: the heavy chain (HC), light intermediate chain (LIC), intermediate chain (IC), and light chain (LC) (Fig. 1a). The HC has both ATPase and MT-binding activities and thus functions as the motor domain[3–5]. The LICs serve as a major binding interface for cargo-adaptor proteins[6]. The ICs interact with dynactin, an activator protein complex that is essential for the conformational change of dynein into an active state, and they serve as a scaffold for the assembly of LCs[7–9]. The LCs comprise the most divergent subunit class, and there are six paralogs reported in mammals, including TCTEX1/3, DYNLL1/2, and roadblock-type-1/2 (DYLRB1/2)[2], but their distinct roles are not fully understood. There have been several roles proposed for LCs that are not mutually exclusive, for example, LCs have been reported to bind to unique subsets of cargo proteins in order to create diversity in cargo-dynein interactions[10,11] and have been shown to ensure the tight dimerization of ICs and thus ensure the high processivity of the dynein complex[12,13]. In addition, LCs have also been suggested to have dynein-independent roles[14–17].

Meiosis is a special form of cell division for gametogenesis, and dynein participates in many aspects of meiosis-specific chromosome dynamics. In prophase I of meiosis, the chromosome ends, i.e., the telomeres, attach to the nuclear envelope and are connected to the cytoplasm in a transmembrane manner[18–22]. Dynein accumulates on the cytoplasmic side of the telomere-nuclear envelope attachment sites and drives the movement of the chromosomes, which is essential for the pairing and

[1]Department of Chemistry and Molecular Biology, University of Gothenburg, SE-41390 Gothenburg, Sweden. [2]Department of Molecular, Cellular, and Developmental Biology, University of Michigan, Ann Arbor, MI 48109, USA. [3]Institute for Quantitative Biosciences, University of Tokyo, 1-1-1 Yayoi, Tokyo 113-0032, Japan. ✉e-mail: hiroki.shibuya@gu.se

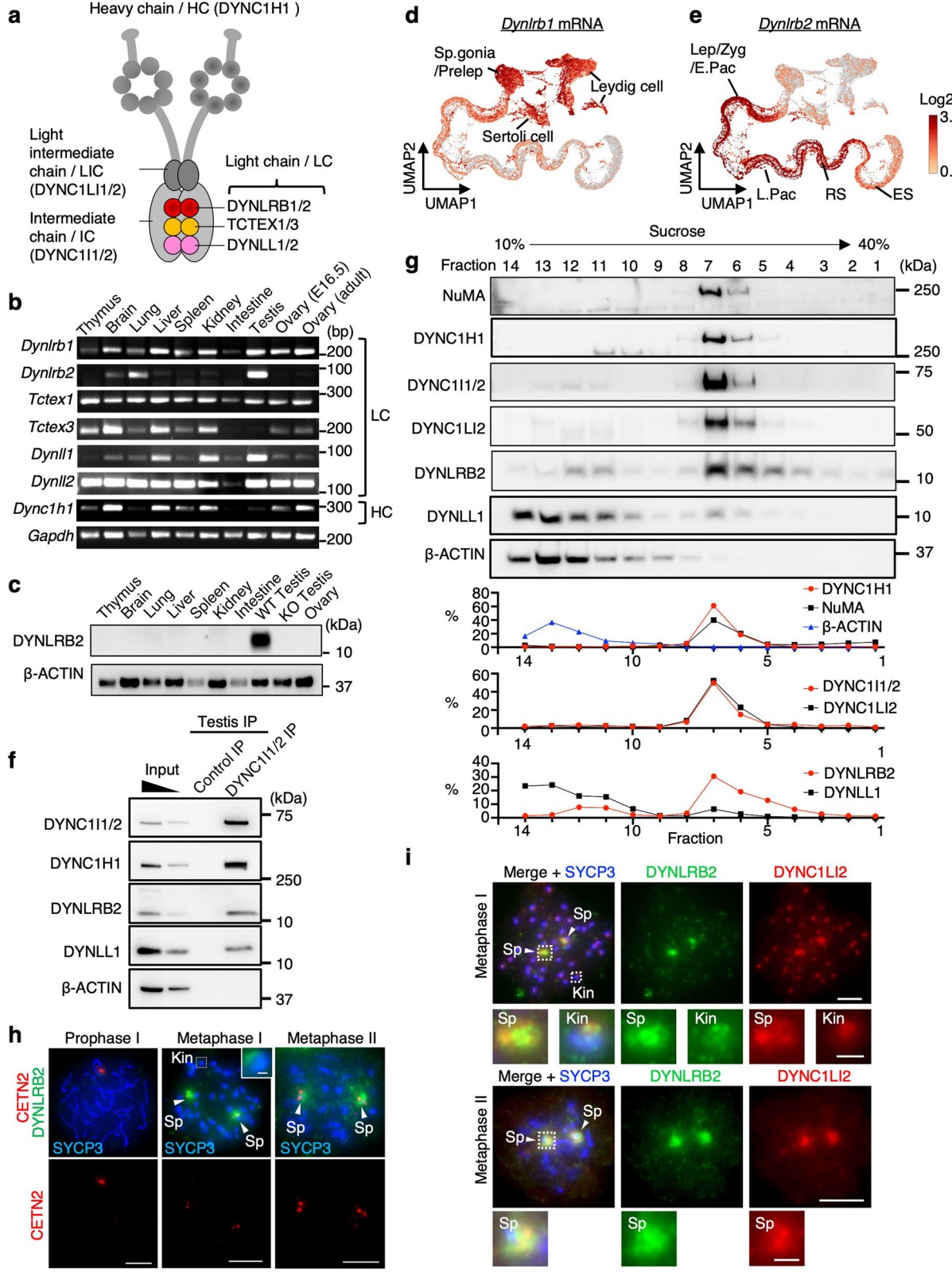

recombination of homologous chromosomes[21,23]. After prophase I, dynein is involved in the assembly of the meiotic spindle, which captures fused sister kinetochores in a monopolar manner different from the bipolar attachment seen in mitosis[24,25]. Even with the unique requirement for these meiosis-specific tasks, the meiosis-specific subunits or modifications of dynein have not been identified.

In this study, we identify DYNLRB2 as a testis-upregulated dynein LC. The expression of DYNLRB2 and its close paralog DYNLRB1 are mutually exclusive and are specific to meiotic and mitotic cells, respectively. The analysis of *Dynlrb2* knockout (KO) mice and *Dynlrb1* knockdown (KD) cells show the specific requirement of DYNLRB1/2 for mitotic and meiotic spindle bipolarity, respectively, by regulating NuMA and centrioles.

**Fig. 1 | Characterization of DYNLRB2 as a testis-upregulated dynein LC.**
**a** Schematic of the cytoplasmic dynein complex 1. **b** Tissue-specific expressions of indicated mouse genes analyzed by reverse transcription PCR. E embryonic day. **c** Tissue-specific expressions of DYNLRB2 and β-ACTIN analyzed by western blotting. The *Dynlrb2⁻/⁻* (KO) testis was used as the negative control. **d** and **e** Uniform manifold approximation and projection (UMAP) representation of *Dynlrb1* (**d**) and *Dynlrb2* (**e**) expression in mouse testes. Multiplexed single-cell RNA sequencing data from PD5−25 WT mouse testis were used. Spermatogonia (Sp.gonia), pre-leptotene (Prelep), leptotene (Lep), zygotene (Zyg), early-pachytene (E.Pac), late-pachytene (L.Pac), round spermatid (RS), elongated spermatid (ES). Expression

patterns of marker genes used for cell-type definitions are shown in Supplementary Fig. 1. **f** IPs from mouse testis extracts with the indicated antibodies. **g** Mouse testis extracts run on a 10−40% sucrose density gradient and analyzed by western blotting. Graphs show the quantification of band intensities with each fraction normalized to the sum of all factions. **h** Immunostaining of WT spermatocytes. Spindle pole (Sp), kinetochore (Kin). The kinetochore signal in metaphase I with intensified DYNLRB2 is shown in the magnified view. Scale bar: 5 μm (1 μm, magnified panel). **i** Immunostaining of WT spermatocytes. Spindle pole (Sp), kinetochore (Kin). The spindle pole signals and kinetochore signals with intensified DYNLRB2 are shown in the magnified view. Scale bar: 5 μm (1 μm, magnified panel).

## Results

### DYNLRB2 is a testis-upregulated dynein LC

To investigate the tissue-specific roles of individual dynein LCs in mice, we examined the mRNA expression of each LC gene by reverse transcription PCR (Fig. 1a). While most of the LC genes were ubiquitously expressed in various somatic tissues along with the HC gene (*Dync1h1*), *Dynlrb2* mRNA was specifically upregulated in testis in addition to moderate expression levels in the brain and lung (Fig. 1b). Western blotting showed the expression of the DYNLRB2 protein exclusively in testes and not in any somatic tissues, suggesting that DYNLRB2 is testis-specific at the protein level in adult mice (Fig. 1c). Interestingly, the closely related paralog *Dynlrb1* was expressed in all the somatic tissues tested (Fig. 1b). The analysis of published single-cell RNA sequencing data from mouse testis[26] showed that the expressions of *Dynlrb1* and *Dynlrb2* were mutually exclusive within the testis with *Dynlrb1* being found in mitotic cells (Sertoli cells, Leydig cells, and spermatogonia) and *Dynlrb2* being found in meiotic cells (Supplementary Fig. 1a, b and Fig. 1d, e). Together these results suggest that *Dynlrb1* and *Dynlrb2* are differentially expressed within mouse tissues and act as mitotic and meiotic counterparts, respectively.

### DYNLRB2 localizes to spindle poles in male meiosis

To test if DYNLRB2 forms a functional dynein complex in the testis, we performed immunoprecipitation (IP) of IC proteins (DYNC1I1/2)[27]. The DYNC1I1/2 IP from mouse testis extracts successfully co-immunoprecipitated DYNLRB2 as well as other dynein subunits (Fig. 1f). The reciprocal DYNLRB2 IP co-immunoprecipitated dynein subunits (Supplementary Fig. 2a), and yeast two-hybrid analyses also confirmed the direct interactions between DYNLRB2 and the ICs (Supplementary Fig. 2b, c). Furthermore, a sucrose density gradient showed that the large majority of DYNLRB2 proteins were co-fractionated into high molecular weight fractions with other dynein subunits, as well as with the dynein-interacting protein NuMA[28], suggesting that DYNLRB2 indeed forms dynein complexes in the testis (Fig. 1g). In contrast to DYNLRB2, another LC protein, DYNLL1, mostly presented as a small dynein-free form (Fig. 1g), which reconciled with the notion that DYNLL1 mainly has dynein-independent roles by acting as a general dimerization hub[14].

To investigate the cell-type-specific expression and localization of DYNLRB2 in the testis, we examined the localization of DYNLRB2 by immunostaining. Even though we could not detect any specific signals in prophase I spermatocytes, there were strong focal signals at the spindle poles in metaphase I and II spermatocytes (Fig. 1h) and these signals co-localized with another dynein subunit, DYNC1LI2 (Fig. 1i). In addition to the spindle pole signals, faint kinetochore signals of DYNLBR2 were detected in metaphase I spermatocytes (Fig. 1h), which was in line with the known kinetochore functions of dynein in mitotic cells[1]. Taken together, these findings lead to the conclusion that DYNLRB2 is a testis-upregulated dynein LC that localizes to spindle poles and kinetochores in meiosis I and II.

### DYNLRB2 is required for male meiosis and fertility

To investigate the cellular functions of DYNLRB2, we generated *Dynlrb2* KO (*Dynlrb2⁻/⁻*) mice (Fig. 2a). In contrast to the reported

embryonic lethality of *Dynlrb1* KO mice[29], *Dynlrb2⁻/⁻* mice were viable. We sometimes observed a smaller body size in juvenile *Dynlrb2⁻/⁻* mice, and these mice developed hydrocephalus (Supplementary Fig. 3a−c). Ciliary dysfunctions caused by depletion of axonemal dynein have been reported to be a cause of hydrocephalus[30], which, together with our observation, suggests that DYNLRB2 might play temporal roles in cilia development in the juvenile brain as a subunit of axonemal dynein. Except for occasional defects in the juvenile brain, the *Dynlrb2⁻/⁻* mice developed into adults without any overt somatic defects. Western blot and immunostaining analyses indicated that DYNLRB2 protein expression was indeed abolished in *Dynlrb2⁻/⁻* mice (Fig. 2b and Supplementary Fig. 3d).

*Dynlrb2⁻/⁻* female mice were fertile (Supplementary Fig. 3e), but *Dynlrb2⁻/⁻* male mice were completely infertile. Consistent with this, the *Dynlrb2⁻/⁻* male mice had smaller testes compared to their WT littermates (Fig. 2c). Histological analysis showed that there were only a small number of post-meiotic spermatids in the *Dynlrb2⁻/⁻* seminiferous tubules, suggesting that the progression of meiosis was highly defective in the mutant testis (Fig. 2d). Consequently, there were no spermatozoa in the *Dynlrb2⁻/⁻* epididymis (Fig. 2d).

To define the specific cell stage at which the mutant cells died, we performed a TdT-mediated dUTP nick-end labeling (TUNEL) assay. The TUNEL assay detected a number of apoptotic cells during spermatocyte development at postnatal day (PD)60, but not during spermatogonia development at PD7, in the *Dynlrb2⁻/⁻* seminiferous tubules (Fig. 2e). In PD60 testes, TUNEL signals were specifically detected in metaphase I spermatocytes from the epithelium of stage XII seminiferous tubules (Fig. 2f). Further, the analysis of testis sections and spermatocyte chromosome spreads demonstrated the significant accumulation of metaphase I spermatocytes within *Dynlrb2⁻/⁻* testes (Fig. 2g, h). These data suggest that DYNLRB2 is required for spermatocyte progression beyond metaphase I.

In meiosis, dynein localizes at telomeres during prophase I and drives the telomere-directed chromosome movements that are essential for homologous pairing and recombination[21,23]. The dynein-dynactin subunits, but not DYNLRB2, were indeed localized at meiotic telomeres during prophase I (Supplementary Fig. 4a−d). In line with the lack of overt defects in prophase I progression prior to metaphase I in the *Dynlrb2⁻/⁻* testis, we confirmed that the telomeric localization of dynein−dynactin and the completion of homologous recombination were completely intact in *Dynlrb2⁻/⁻* spermatocytes (Supplementary Fig. 4a−c, e), indicating that DYNLRB2 is dispensable for prophase I-specific dynein function but plays specific roles in metaphase I.

### Spindle defects in *Dynlrb2⁻/⁻* metaphase I spermatocytes

To study the cellular defects in *Dynlrb2⁻/⁻* metaphase I spermatocytes, we examined the meiotic spindle structure by staining for α-tubulin together with SYCP3, which accumulates at centromeres in metaphase I. Most of the WT spermatocytes formed normal bipolar spindles, but the majority of *Dynlrb2⁻/⁻* metaphase I spermatocytes had more than two spindle poles thus forming multipolar spindles (Fig. 3a, b). Even spindles with two apparent spindle poles had various structural defects such as abnormally widened, elongated, and misoriented spindles (Fig. 3a, c). In line with the various spindle defects,

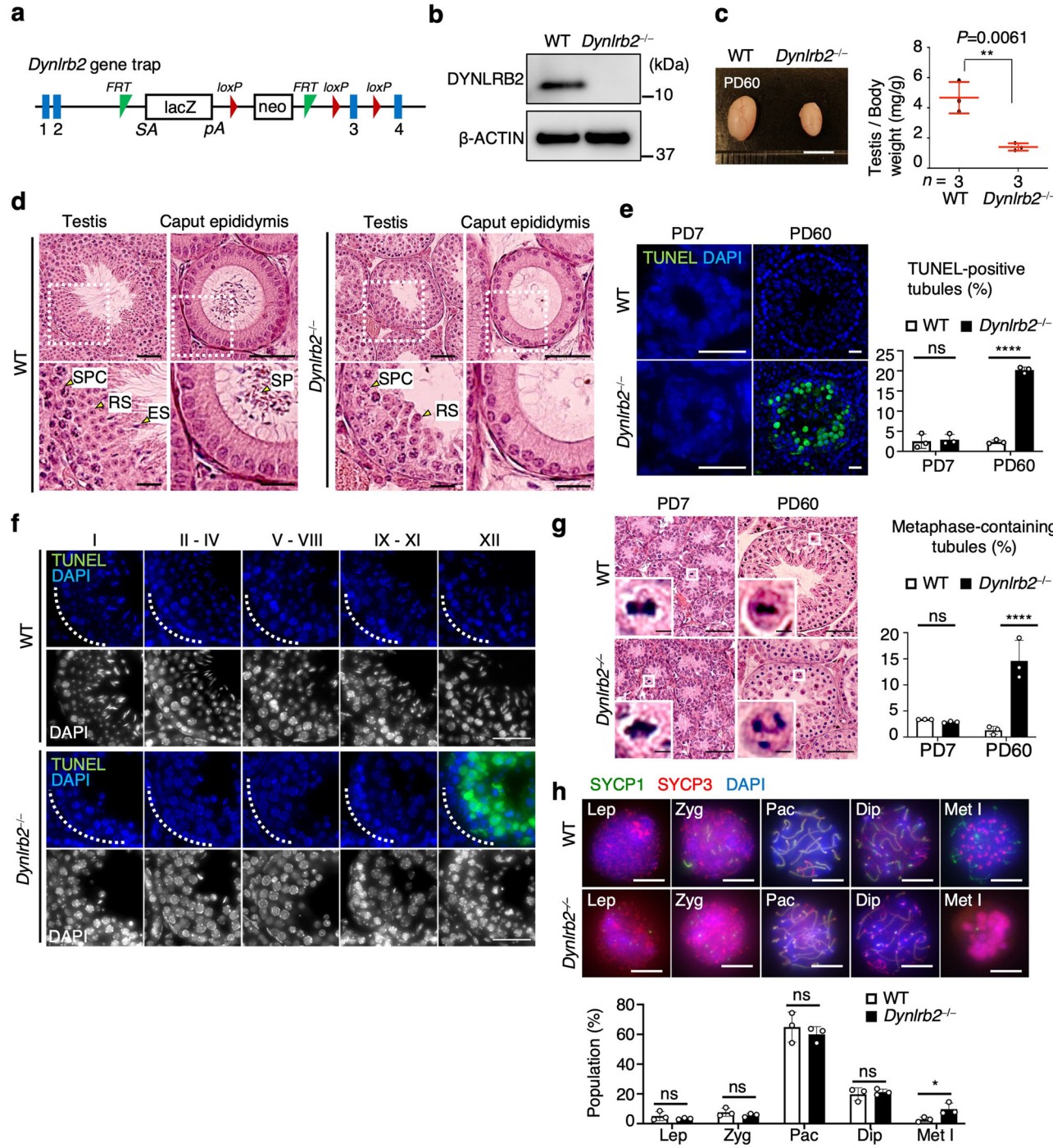

**Fig. 2 | DYNLRB2 is required for male meiosis progression and fertility. a** The *Dynlrb2* gene trap allele. Rectangles represent exons. Flippase recognition target; *FRT*. Neomycin resistance cassette; *Neo*. Splicing accepter; *SA* and Polyadenylation signal; *pA*. **b** Western blotting of mouse testis extracts from WT and *Dynlrb2⁻/⁻* mice. **c** Testes from PD60 mice and the mean values with SD of the testis to body weight ratio using three different mice. Scale bar: 5 mm. **d** PD60 testis and epididymis sections stained with hematoxylin and eosin. Spermatocyte (SPC), round spermatid (RS), elongated spermatid (ES), spermatozoon (SP). Scale bar: 50 μm (20 μm, magnified panel). Spermatozoa were observed in 99.8% (*n* = 5114 tubules) and 0% (*n* = 3828 tubules) of epididymal tubule sections in WT and *Dynlrb2⁻/⁻* epididymides, respectively. **e** Testis sections at PD7 and PD60 stained with TUNEL and DAPI. The percentages of TUNEL-positive seminiferous tubules (those containing more than three TUNEL-positive cells) were quantified. More than 700 seminiferous tubules were quantified for each condition. The mean values with SD of three independent experiments are shown. Scale bar: 10 μm at PD7 and 20 μm at PD60. **f** Testis

sections at PD60 stained with TUNEL and DAPI. Scale bar: 10 μm. **g** Testis sections at PD7 and PD60 stained with hematoxylin and eosin. The percentages of seminiferous tubules containing mitotic metaphase cells and meiotic metaphase I spermatocytes were quantified at PD7 and PD60, respectively. More than 1800 seminiferous tubules were quantified for each condition. The mean values with SD of three independent experiments are shown. Scale bar: 50 μm (5 μm, magnified panel). **h** Spermatocyte chromosome spreads stained with the indicated antibodies and DAPI. SYCP3-positive spermatocytes (1172 cells for WT and 1390 cells for KO) were classified into the following substages: Lep leptotene (no SYCP1); Zyg zygotene (partially assembled SYCP1); Pac pachytene (fully assembled SYCP1); Dip diplotene (disassembled SYCP1); and Met I metaphase I (SYCP3 at centromeres and no SYCP1). The mean values with SD of three independent experiments are shown. Scale bar: 5 μm. All analyses used two-tailed *t*-tests. ns: not significant, *\*p* < 0.05, *\*\*p* < 0.01, *\*\*\*\*p* < 0.0001.

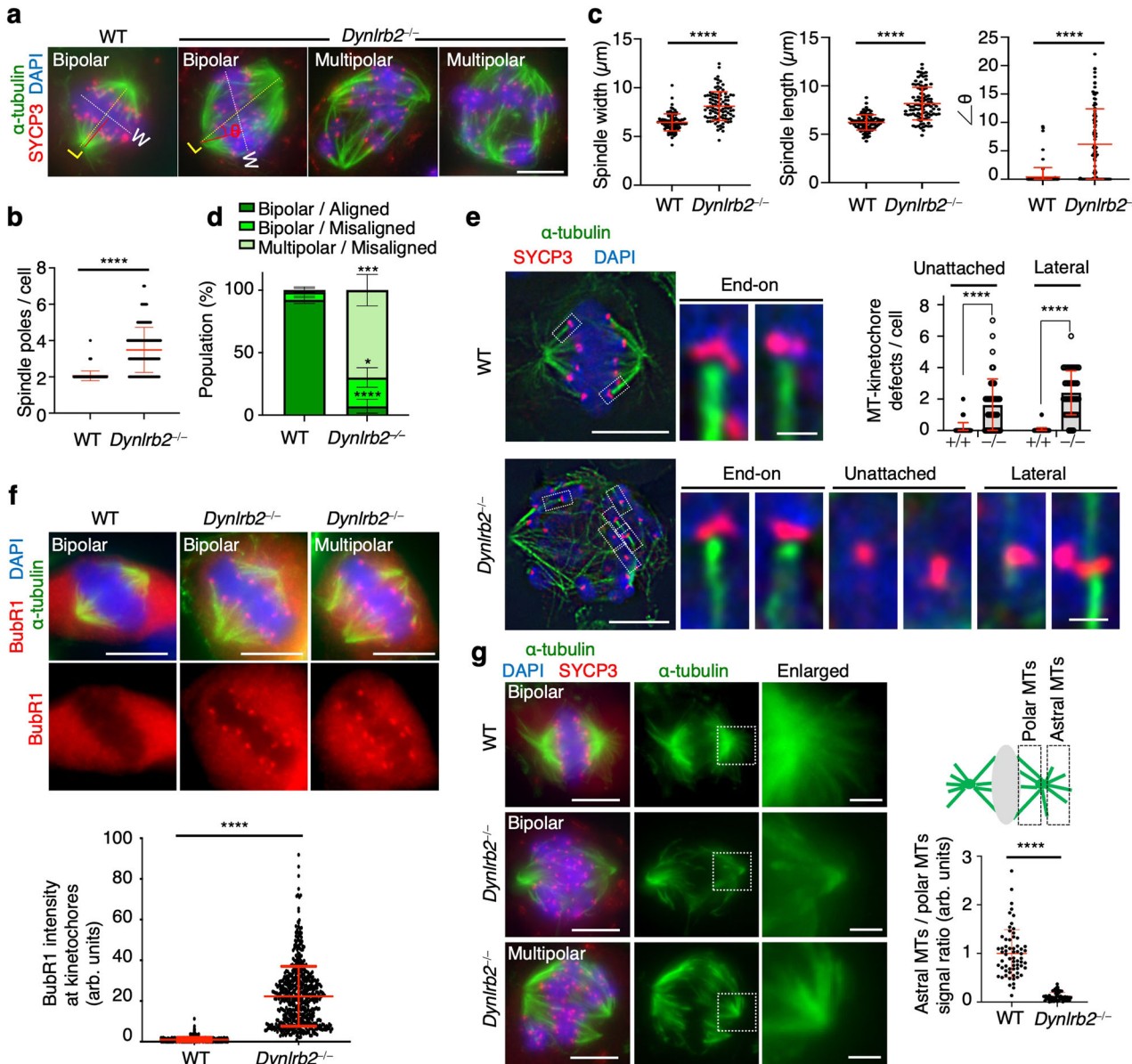

**Fig. 3 | Spindle defects in *Dynlrb2* KO spermatocytes. a** Immunostaining of metaphase I spermatocytes prepared by the squash technique. The length between two spindle poles (L), the width of the metaphase plate (W), and the angle between a line perpendicular to the metaphase plate and a line connecting the two spindle poles ($\theta$) are indicated. Scale bar: 5 μm. **b** The quantification of spindle pole number from 118 metaphase I spermatocytes collected from three different mice for each genotype. Mean values with SD are shown. **c** The quantification of spindle width ($n = 98$ cells for each genotype), length ($n = 98$ cells for each genotype), and angle ($\theta$) ($n = 100$ cells for each genotype). Mean values with SD are shown. Only cells that formed bipolar spindles were quantified. **d** The mean values with SD of chromosome misalignment from three independent experiments ($n = 118$ cells and 193 cells for WT and *Dynlrb2−/−*, respectively). **e** Immunostaining of metaphase I spermatocytes prepared by the squash technique. The mean values with SD of the frequency of kinetochore-MT attachment errors per metaphase I spermatocytes from three independent experiments are shown ($n = 45$ cells for both WT and *Dynlrb2−/−*). Scale bar: 5 μm (0.5 μm, magnified panel). **f** Immunostaining of spermatocytes prepared by the squash technique. The BubR1 signal intensities at metaphase I kinetochores were quantified and normalized by the average value of the WT controls. The mean values with SD from three independent experiments are shown ($n = 580$ collected from 30 metaphase I cells for both WT and *Dynlrb2−/−*). Scale bar: 5 μm. **g** Immunostaining of metaphase I spermatocytes prepared by the squash technique. The mean values with SD of the quantification of astral MT intensity normalized by the polar MT intensity from three independent stainings for each genotype are shown. The mean values with SD from three independent stainings for each genotype are shown ($n = 64$ spindle poles from 32 cells for both WT and *Dynlrb2−/−*). Only cells that formed bipolar spindles were quantified. Scale bar: 5 μm (1 μm, magnified panel). All analyses used two-tailed *t*-tests. \**p* < 0.05, \*\*\**p* < 0.001, \*\*\*\**p* < 0.0001.

chromosomes were largely misaligned (Fig. 3d). Further, the end-on attachment of kinetochore-MTs was largely impaired (Fig. 3e), and thus the cells activated the spindle assembly checkpoint as indicated by the significant accumulation of the checkpoint component BubR1 at kinetochores (Fig. 3f).

In mitosis, dynein is recruited to the cell cortex and generates pulling forces on the astral microtubules for the proper positioning of

miotic spindles[31]. In the *Dynlrb2−/−* metaphase I spindle, the astral microtubules were completely depleted (Fig. 3g). These data suggest that DYNLRB2 has roles in the stabilization of astral microtubules likely by anchoring them to the meiotic cell cortex.

We next tested if the observed defects were meiosis-specific by examining mitotic cells in the *Dynlrb2−/−* mice. Both embryonic fibroblasts and ear fibroblasts derived from *Dynlrb2−/−* mice showed normal

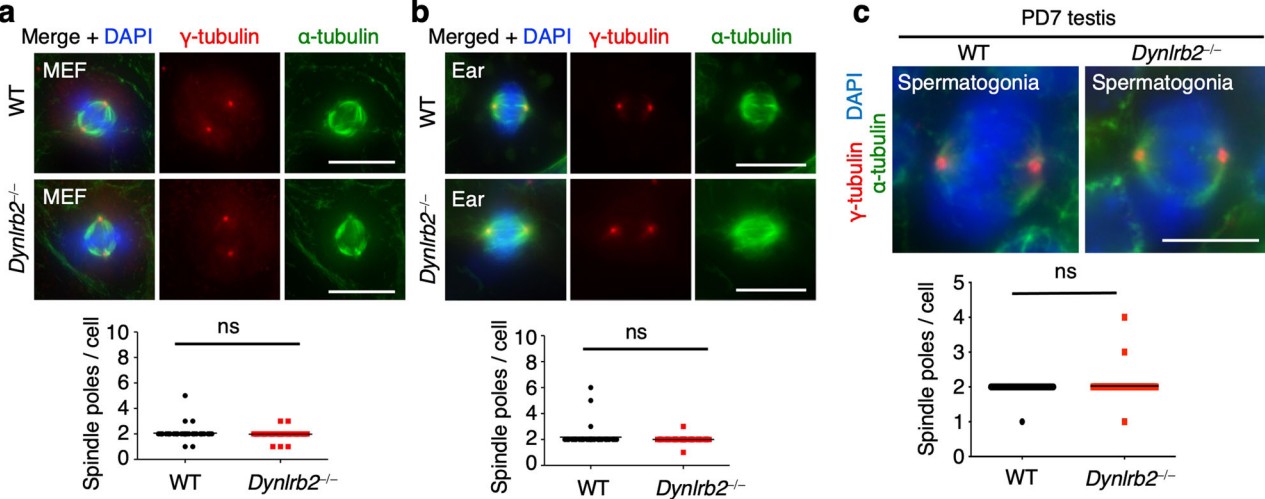

**Fig. 4 | Mitotic spindles are intact in *Dynlrb2* KO mice. a** and **b** Immunostaining of mitotic metaphase cells from mouse embryonic fibroblasts (MEF) (**a**) or ear fibroblasts (Ear) (**b**). The mean values with SD of the number of spindle poles from three independent experiments are shown (*n* = 45 metaphase cells for all conditions). Scale bar: 10 μm. **c** Immunostaining of metaphase spermatogonia from PD7 testes. The mean values with SD of the number of spindle poles from three independent experiments are shown (*n* = 40 metaphase cells for each genotype). Scale bar: 5 μm. All analyses used two-tailed *t*-tests. ns not significant.

mitosis progression with the formation of normal bipolar spindles (Fig. 4a, b). Further, the mitotically rounding spermatogonia in PD7 *Dynlrb2*⁻/⁻ testes also had normal bipolar spindles (Fig. 4c). These data suggest that the spindle formation defects observed in *Dynlrb2*⁻/⁻ mice are specific to male meiosis I.

**Premature centriole disengagement in the absence of DYNLRB2**
To gain mechanical insights into the observed spindle defects, we examined the localization of the PCM proteins γ-tubulin and pericentrin (PCNT)[32]. In WT mice, both γ-tubulin and PCNT formed two distinct foci representing two centrosomes at each spindle pole (Fig. 5a). However, in *Dynlrb2*⁻/⁻ spermatocytes both γ-tubulin and PCNT were frequently fragmented and formed more than two foci (Fig. 5a). The fragmented PCM indeed emanated microtubules and thus formed multipolar spindles (Fig. 5b).

The PCM fragmentation can be attributed to centriolar dysfunctions, i.e., to overduplicated centrioles or to prematurely separated mother and daughter centrioles[33]. In meiotic cells, centrioles are duplicated at pre-leptotene to leptotene stages at the beginning of prophase I (Fig. 5c)[34]. The quantification of the foci number of the centriolar protein CENTRIN2 (CETN2) showed normal kinetics for meiotic centriole duplication in *Dynlrb2*⁻/⁻ spermatocytes (Fig. 5d). Indeed, PCMs were not fragmented and formed tightly paired foci representing the duplicated centrosomes throughout prophase I in *Dynlrb2*⁻/⁻ spermatocytes similar to WT spermatocytes (Supplementary Fig. 5a).

Consistent with the occurrence of normal centriole duplication during prophase I, the four CETN2 signals were observed in *Dynlrb2*⁻/⁻ metaphase I cell (Fig. 5e, f). However, around 23% of the *Dynlrb2*⁻/⁻ metaphase I cells showed premature separation of mother and daughter centrioles (Fig. 5e, g). The prematurely separated centrioles accompanied PCNT signals (Fig. 5h), suggesting that centriole separation led to the PCM fragmentation seen in *Dynlrb2*⁻/⁻ metaphase I spermatocytes.

In mitosis, the mother and daughter centrioles are tightly linked, known as centriole engagement, until the exit from mitosis[35]. Prolonged miotic arrest induces premature centriole disengagement, which is characterized by the premature loss of procentriole assembly factors such as STIL and SAS-6 from the daughter centriole via active proteolysis[36]. To test if premature centriole disengagement could account for the premature separation of mother and daughter

centrioles in *Dynlrb2*⁻/⁻ metaphase I cell, we examined the localization of STIL and SAS-6. In line with the established knowledge from mitotic studies[37], each centriolar pair in the WT metaphase I spermatocytes accompanied single STIL and SAS-6 signals, which corresponded reasonably well to the proximal ends of daughter centrioles (Fig. 5i and Supplementary Fig. 5b, top). However, in *Dynlrb2*⁻/⁻ metaphase I cell with separated centrioles, STIL and SAS-6 signals were completely missing (Fig. 5i and Supplementary Fig. 5b, bottom). In some cases, even paired centrioles were devoid of STIL and SAS-6 signals (Fig. 5i and Supplementary Fig. 5b, second from bottom), suggesting that premature disengagement precedes the separation of mother and daughter centrioles. Taken together, we concluded that DYNLRB2 ensures centriole engagement in metaphase I in order to maintain spindle bipolarity.

**DYNLRB2 enriches NuMA at spindle poles in male meiosis I**
Even though premature centriole disengagement accounts for some of the multipolarity seen in *Dynlrb2*⁻/⁻ spermatocytes, the majority of the multipolar spindles have intact pairs of centrioles (Fig. 5e, g), implying that there are some other pathways through which DYNLRB2 prevents PCM fragmentation (Fig. 5j).

To identify the additional mechanism, we screened for proteins whose localization is specifically affected in *Dynlrb2*⁻/⁻ spermatocytes. Spindle bipolarity is not only ensured by spindle pole proteins, but also by kinetochore-binding proteins, which maintain the stability of the kinetochore−MT interaction and prevent spindle pole collapse due to the continuous shortening of kinetochore−MTs (Fig. 6a)[38,39]. The NDC80 complex and kinesin CENP-E are two major kinetochore proteins responsible for the direct kinetochore−MT attachment[40] and whose depletion causes spindle pole collapse in both mitotic cells and in metaphase I oocytes[41–43]. The immunostaining of NDC80 and CENP-E confirmed that these proteins localized similarly at metaphase I kinetochores in both WT and *Dynlrb2*⁻/⁻ spermatocytes suggesting that the kinetochore−MT interfaces were unaffected (Fig. 6b, c). Meiosis-specific monopolar attachment is ensured by the recruitment of PLK1 to kinetochores via the meiosis-specific kinetochore protein MEIKIN[44], and this was also unaffected in *Dynlrb2*⁻/⁻ spermatocytes (Fig. 6d). These data suggest that the protein composition of the kinetochore, especially the proteins involved in kinetochore−MT attachment and meiosis-specific monopolar attachment, was unaffected in *Dynlrb2*⁻/⁻ spermatocytes.

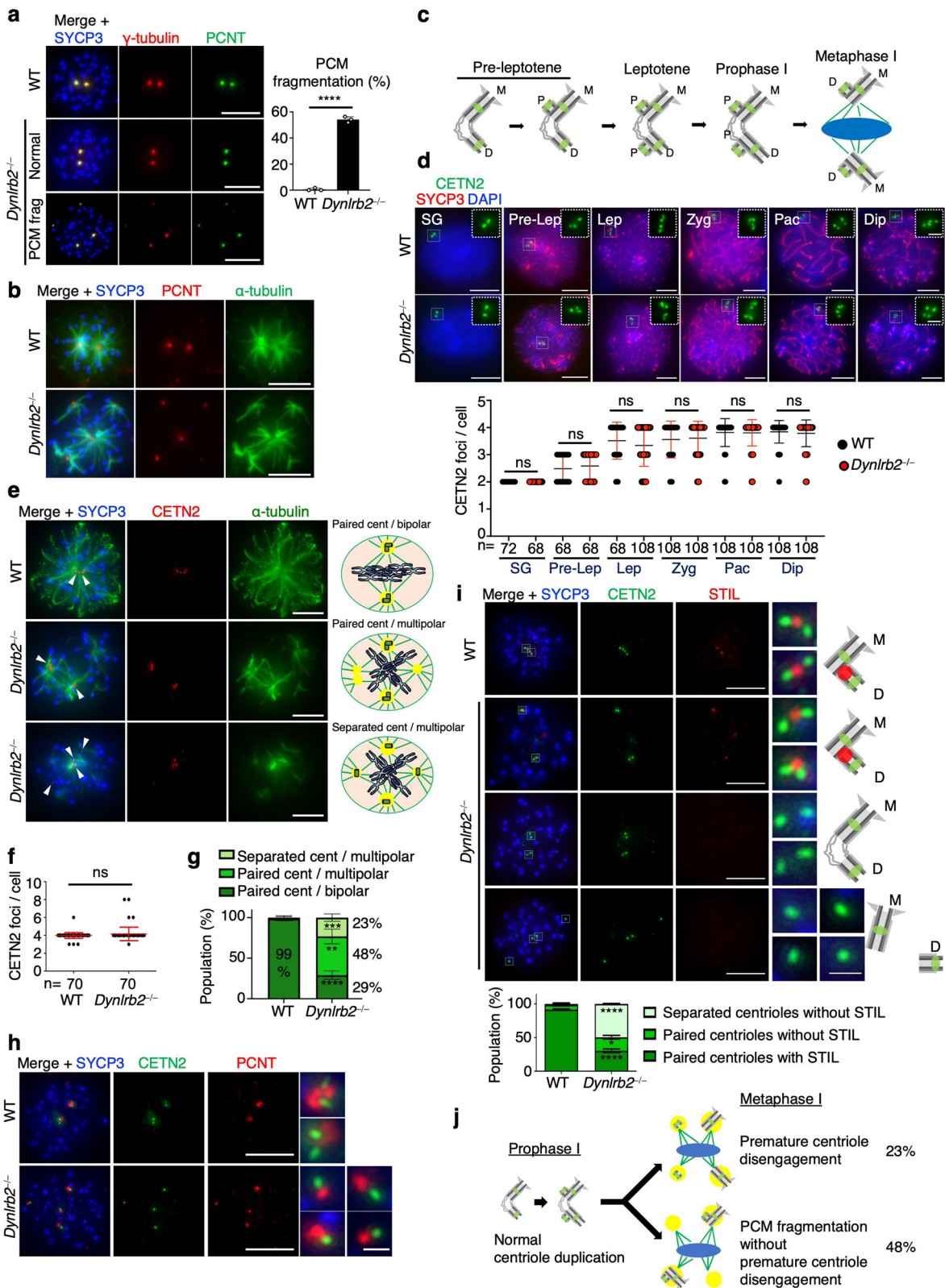

We next examined the localization of spindle pole proteins (Fig. 6a). Sliding of the antiparallel overlap of MTs by the kinesin-5 motor Eg5 is critical for the formation of the bipolar spindle[45], and such sliding is counteracted by dynein–dynactin[46]. We found that the localization of Eg5 at spindle poles and spindle MTs was unaffected in *Dynlrb2*[-/-] spermatocytes (Fig. 6e). To our surprise, the localization of the dynein–dynactin subunits DYNC1H1, DYNLL1, and p150 at spindle poles were not affected in *Dynlrb2*[-/-] spermatocytes, suggesting that dynein complex formation and its retrograde motor activity are not affected by the loss of the DYNLRB2 subunit (Fig. 6f–h). In line with these cytological findings, the IP of DYNC1I1/2 from WT and *Dynlrb2*[-/-] testis extracts confirmed the similar co-IP of other dynein subunits, thus suggesting that dynein complex formation was not impaired in the absence of DYNLRB2 (Fig. 6i).

**Fig. 5 | Premature disengagement of mother and daughter centrioles in the absence of DYNLRB2. a** Immunostaining of metaphase I spermatocytes. The frequency of cells with fragmented PCM were quantified. The mean values with SD of three independent experiments using three different mice are shown ($n$ = 102 and 176 cells for WT and $Dynlrb2^{-/-}$, respectively). Scale bar: 5 μm. **b** Immunostaining of metaphase I spermatocytes. Scale bar: 5 μm. **c** Schematic of centriole duplication in meiotic prophase I. M; mother centriole, D; daughter centriole, and P; procentrioles. **d** Immunostaining of spermatogonia (SG) and spermatocytes at preleptotene (Pre-lep), leptotene (Lep), zygotene (Zyg), pachytene (Pac), and diplotene (Dip). The graph shows the mean CETN2 foci number with SD. n shows the total cell number pooled from three independent experiments using three different mice. Scale bar: 5 μm (0.5 μm, magnified panel). **e** Immunostaining of metaphase I spermatocytes. Arrowheads indicate CETN2 foci. Scale bar: 5 μm. **f** The mean CETN2 foci number with SD at metaphase I. n shows the total cell number pooled from three independent experiments using three different mice. **g** The quantification of centriole and spindle defects. The mean values with SD of three independent experiments using three different mice are shown ($n$ = 77 and 109 cells for WT and $Dynlrb2^{-/-}$, respectively). **h** Immunostaining of metaphase I spermatocytes. Scale bar: 5 μm (0.5 μm, magnified panel). **i** Immunostaining of metaphase I spermatocytes. The graph shows the quantification of centriolar defects. The mean values with SD of three independent experiments using three different mice are shown ($n$ = 59 and 147 cells for WT and $Dynlrb2^{-/-}$, respectively). Scale bar: 5 μm (0.5 μm, magnified panel). **j** Schematic of centriole and spindle defects in $Dynlrb2^{-/-}$ spermatocytes. All analyses used two-tailed $t$-tests. ns: not significant, \*\*$p < 0.01$, \*\*\*$p < 0.001$, \*\*\*\*$p < 0.0001$.

We next examined the localization of NuMA, a dynein-binding protein that accumulates at the spindle pole[28]. NuMA has an intrinsic MT minus-end binding activity that localizes it to the spindle pole and ensures spindle bipolarity[47]. It is widely accepted that NuMA accumulates at spindle poles in a dynein-independent manner and recruits dynein to the spindle pole[47,48], although there are some reports suggesting the reverse hierarchy[49]. All of these studies are based on observations in mitotic cells, and the regulatory mechanisms in male meiotic spindles have not been addressed. As expected from studies in mitotic cells, NuMA localized on spindles with strong focal signals at their spindle poles in WT spermatocytes (Fig. 6j, top). However, to our surprise NuMA was largely mislocalized from spindles in $Dynlrb2^{-/-}$ spermatocytes (Fig. 6j) or in some cases was bound to spindles but failed to concentrate at spindle poles (Supplementary Fig. 6). The signal quantification showed a 60% reduction in NuMA intensity in $Dynlrb2^{-/-}$ spermatocytes compared to the WT controls (Fig. 6k). The line scan analysis also showed that the symmetric distribution of NuMA at each spindle pole seen in WT was largely disrupted in $Dynlrb2^{-/-}$ spermatocytes (Fig. 6l), suggesting that the poleward concentration of NuMA was impaired. Western blotting analysis confirmed that the expression of NuMA protein was unaffected in the mutant testis (Fig. 6m). Thus we conclude that DYNLRB2 is crucial for the proper recruitment of NuMA to meiotic spindle poles and for the maintenance of spindle pole integrity.

## DYNLRB1 ensures spindle pole integrity in mitotic cells

DYNLRB1 and DYNLRB2 share 75% amino acid identity (Supplementary Fig. 7a), thus raising the possibility that DYNLRB1 has overlapping roles with meiotic DYNLRB2 in mitotic cells. We used the mouse B16-F1 melanoma cell line to investigate the mitotic role of DYNLRB1. In line with the hypothesis that DYNLRB1/2 are mitotic and meiotic counterparts, respectively, only $Dynlrb1$ and not $Dynlrb2$ was expressed in B16-F1 cells (Fig. 7a). The transfection of $Dynlrb1$ siRNA efficiently knocked down the expression of $Dynlrb1$ (Fig. 7b), which caused significant mitotic arrest (Fig. 7c). As expected, most of the metaphase cells in the $Dynlrb1$ knockdown (KD) group had multipolar spindles and fragmented PCMs (Fig. 7d). Accordingly, the chromosomes were largely misaligned (Fig. 7e). We observed occasional abnormal telophase cells interconnected by more than one mid-body and interphase cells interconnected by long threads of cytoplasm, both of which were likely caused by mitotic slippage after the formation of multipolar spindles (Fig. 7f).

We next examined if centrioles were misregulated in $Dynlrb1$ KD cells. Different from the $Dynlrb2^{-/-}$ meiotic cells, $Dynlrb1$ KD cells with fragmented PCM had more than four CETN2 foci (Fig. 7g), and the quantification showed that the fragmented PCM was mostly accompanied by CETN2 foci (Fig. 7h), suggesting that the PCM fragmentation was primarily caused by the overduplication of centrioles in $Dynlrb1$ KD cells. The presence of supernumerary centrioles was reported in DYNC1LI1/2-deficient cells as well[50], and these defects can also be attributed to cytokinesis errors in the previous cell cycle (Fig. 7f).

To determine if NuMA was misregulated as it was in $Dynlrb2$ KO spermatocytes, we examined the localization of NuMA in $Dynlrb1$ KD cells. Even though the reduction of the signal intensity was less drastic compared to the $Dynlrb2$ KO spermatocytes, we still detected a significant reduction in NuMA signal intensity (62% of the control level), suggesting that DYNLRB1 ensures the robust accumulation of NuMA at mitotic spindle poles (Fig. 7i, j). Previous studies in human mitotic cell lines concluded that NuMA localization at spindle poles is dynein-independent[47,48], which does not reconcile with our observations. To rule out the possibility that our findings were cell-type or species-specific, we knocked down human $DYNLRB1$ (h$DYNLRB1$) in HEK293 cells. We confirmed that all of the defects seen in murine B16-F1 cells were recapitulated in h$DYNLRB1$ KD HEK293 cells, suggesting that DYNLRB1's roles in mitosis are conserved across species (Supplementary Fig. 7b–g).

## Ectopic DYNLRB2 expression rescues the $Dynlrb1$ KD phenotypes

Finally, we sought to determine if the mitotic role of DYNLRB1 can be replaced by DYNLRB2 in an RNAi rescue experiment in B16-F1 cells and HEK293 cells. The expression of RNAi-resistant $^{GFP}$DYNLRB1 completely restored the multipolarity caused by $Dynlrb1$ KD (Fig. 8a and Supplementary Fig. 7h). Similarly, the expression of $^{GFP}$DYNLRB2 restored spindle multipolarity (Fig. 8a and Supplementary Fig. 7h), thus demonstrating that the mitotic DYNLRB1 functions are complemented by the ectopically expressed DYNLRB2. To determine the motor activities of the distinct dynein complexes, we transfected either $^{FLAG}$hDYNLRB1 or $^{FLAG}$hDYNLRB2 into HeLa cells stably expressing GFP-labeled dynein HC and purified the active GFP–dynein complexes by FLAG-IP (Fig. 8b). The following single-molecular analyses demonstrated that both dynein complexes moved processively along the in vitro-polymerized MTs with comparable velocities (Fig. 8c and Supplementary movies 1, 2), supporting the notion that hDYNLRB1 and hDYNLRB2 have interchangeable roles with respect to the motility of the mitotic dynein complex.

## Discussion

Our results call for the reinterpretation of previous works. A study using HeLa cells and Xenopus egg extracts reported that the spindle pole accumulation of NuMA is dependent on the dynein/dynactin-dependent transport of NuMA toward spindle poles[49]. However, several follow-up studies using human cell lines, including HeLa cells and PtK2 cells, showed that NuMA localizes to spindle poles independent of dynein and that this in turn recruits dynein to the spindle poles[47,48]. The latter concept is now widely accepted in the research field. The analysis of $Dynlrb2$ KO mice and $Dynlrb1$ KD cell lines reported here showed that NuMA localization at spindle poles mostly depends on the meiosis-specific DYNLRB2-containing dynein complex in male meiosis I and partially depends on the mitotic counterpart, the DYNLRB1-containing dynein complex, in both murine and human mitotic cell lines. This discrepancy between the preceding studies and our results

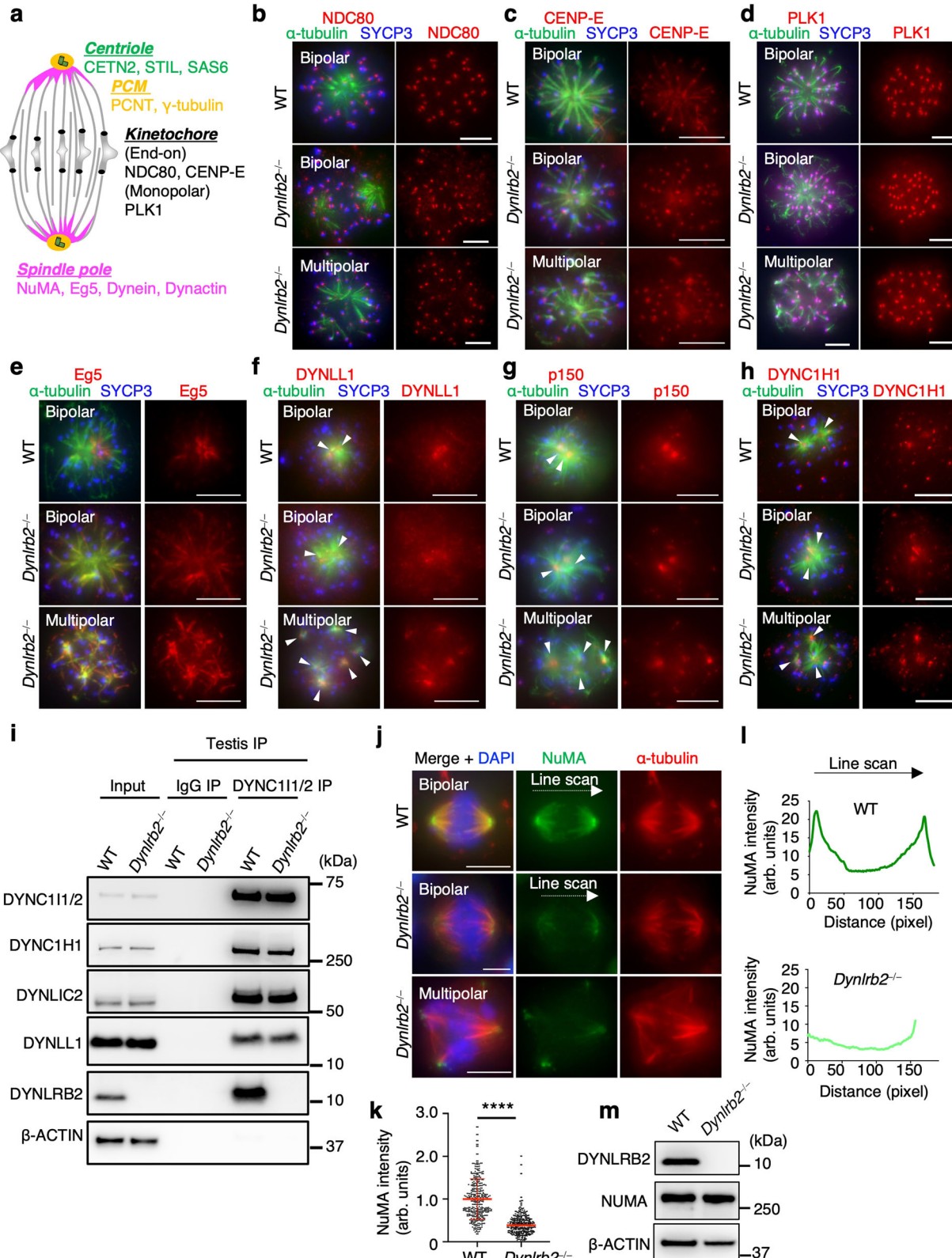

**Fig. 6 | DYNLRB2 recruits NuMA to spindle poles to ensure MT focusing. a** Schematic of meiotic spindle structure with major proteins that contribute to spindle bipolarity. **b–h** Immunostaining of metaphase I spermatocytes with indicated antibodies. The arrowheads indicate spindle poles. Scale bar: 5 μm. **i** IPs from WT (+/+) and *Dynlrb2* KO (−/−) mouse testis extracts with the indicated antibodies. **j** Immunostaining of metaphase I spermatocytes prepared by the squash technique. Scale bar: 5 μm. **k** The quantification of NuMA intensity at spindle poles normalized by the average value of the WT controls. The mean values with SD are shown. Data were pooled from five independent stainings using four different mice for each genotype (*n* = 194 spindle poles from 97 cells for both WT and *Dynlrb2*−/−). Only cells that formed bipolar spindles were quantified. The analysis used two-tailed *t*-tests. ****$p < 0.0001$. **l** The line scan analysis of NuMA intensity between each spindle pole. **m** Mouse testis extract analyzed by western blotting with the indicated antibodies.

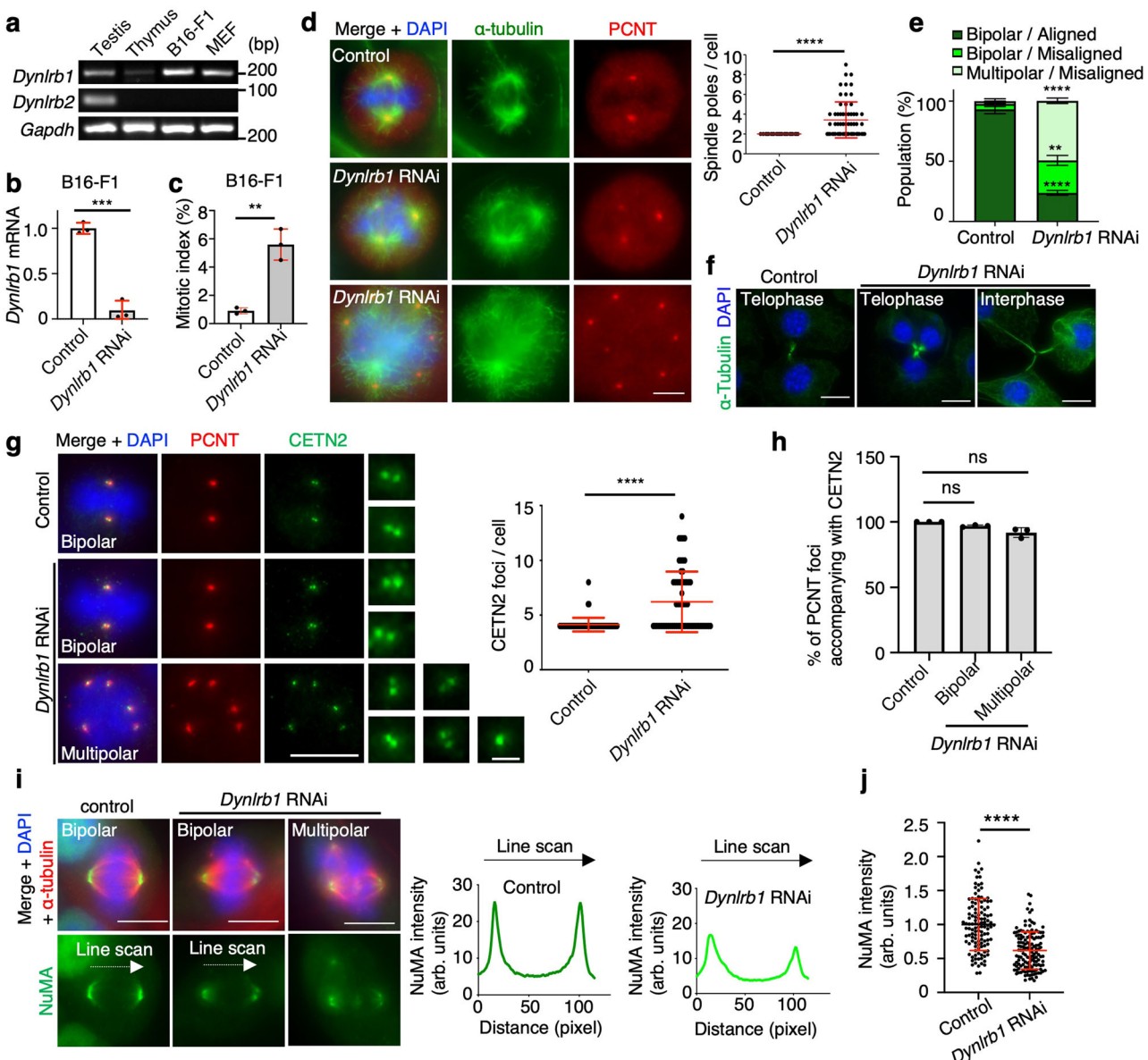

**Fig. 7 | DYNLRB1 ensures spindle pole integrity in mitosis. a** Tissue and cell-type-specific expression of the indicated mouse genes. **b** The quantification of the *Dynlrb1* mRNA expression level normalized to the average value of the controls in B16-F1 cells. The mean values with SD of three independent experiments are shown. **c** The quantification of mitotic cell population in B16-F1 cells. The mean values with SD of three independent experiments are shown ($n = 2275$ and $2058$ for control and *Dynlrb1* RNAi, respectively). **d** Immunostaining of B16-F1 cells in metaphase and the number of spindle poles. Mean values with SD from three independent experiments for both conditions are shown ($n = 65$ for both conditions). Scale bar: 5 µm. **e** Quantification of chromosome alignment and spindle defects in B16-F1 cells. The mean values with SD of three independent experiments are shown ($n = 90$ and $209$ for control and *Dynlrb1* RNAi, respectively). **f** Immunostaining of B16-F1 cells. Scale bar: 10 µm. **g** Immunostaining of B16-F1 cells in metaphase and quantification of CETN2 foci number from three independent experiments for both conditions

($n = 43$ for both conditions). Scale bar: 5 µm (0.5 µm, magnified panel). **h** The quantification of PCNT foci accompanying CETN2 foci in B16-F1 metaphase cells. The mean values with SD of three independent experiments are shown. PCNT foci ($n = 190$, $104$, and $196$) were pooled from $95$, $49$, and $46$ cells for control, *Dynlrb1* RNAi (bipolar), and *Dynlrb1* RNAi (multipolar), respectively. **i** Immunostaining of B16-F1 cells in metaphase. The graphs show the line scan analysis of NuMA intensity between each spindle pole. Scale bar: 5 µm. **j** The quantification of NuMA signal intensity at metaphase spindle poles in B16-F1 cells normalized to the average value of the controls. The mean values with SD from three independent experiments are shown ($n = 118$ and $138$ for control and *Dynlrb1* RNAi, respectively). Only cells that formed bipolar spindles were quantified. Two-tailed *t*-tests were used in (**b**–**d**), (**e**), (**g**), and (**j**). One-way ANOVA with Dunnett's multiple comparisons test was used in (**h**). ns not significant, **$p < 0.01$, ***$p < 0.001$, ****$p < 0.0001$.

can be partly explained by assuming the presence of secondary binding sites between dynein and NuMA. The direct interaction between NuMA and the LIC subunits of dynein has been considered to be the sole binding site between NuMA and dynein, but the loss of this interaction by introducing point mutations in the LIC did not abrogate the NuMA localization at spindle poles[48]. Because NuMA is a very large protein with a molecular weight of 238 kilodaltons, which prevents the

purification and analysis of full-length protein in vitro, the previous in vitro studies used the short N-terminus fragment of NuMA that contains a Hook domain that mediates the LIC interaction. The C-terminus extension of NuMA might have secondary binding sites for dynein, and DYNLRB1/2 or some other protein or proteins recruited to dynein by DYNLRB1/2 might be the corresponding binding interface within the dynein complex. This secondary interaction together with

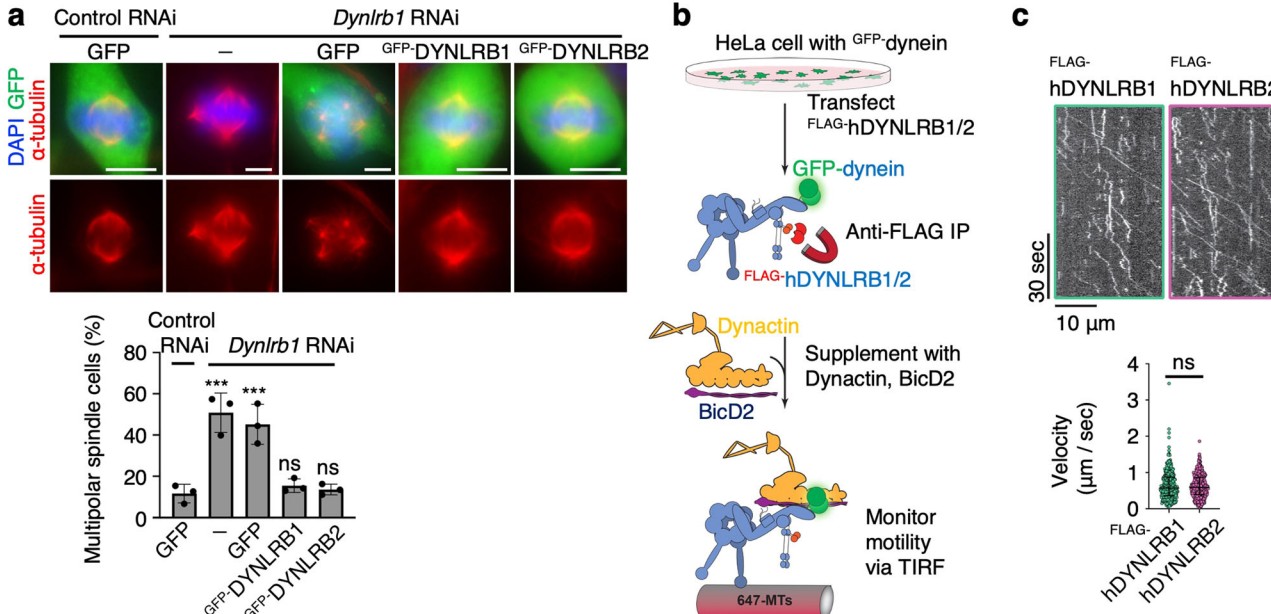

**Fig. 8 | DYNLRB1 and DYNLRB2 have interchangeable roles in mitosis.**
**a** Immunostaining of B16-F1 cells in metaphase and the frequency of metaphase cells that formed multipolar spindles. The mean values with SD of three independent experiments are shown (more than 300 metaphase cells were quantified for each condition). Scale bar: 5 µm. **b** Schematic of single-molecular analysis for purified GFP–dynein complexes. **c** Example kymographs showing processive

events for dynein immunoprecipitated with either [FLAG]hDYNLRB1 or [FLAG]hDYNLRB2. The graph shows the velocity measurements of processive events from three biological replicates. Each dot represents an individual event ($n = 463$ and 385 for [FLAG]hDYNLRB1 and [FLAG]hDYNLRB2, respectively). Medians with interquartile ranges are shown. Two-tailed $t$-tests were used in (**c**). One-way ANOVA with Dunnett's multiple comparisons test was used in (**a**). ns not significant, \*\*\*$p < 0.001$.

the known NuMA–LIC interaction might synergistically reinforce the NuMA–dynein association to ensure the robust accumulation of NuMA via dynein-dependent retrograde transport toward spindle poles.

The key phenotype seen in *Dynlrb2* KO metaphase I spermatocytes is the fragmentation of PCM and the formation of multipolar spindles, which we found were caused by two distinct mechanisms; i.e., premature centriole disengagement and NuMA mislocalization. In mitosis, centriole engagement is maintained by the cohesin complex localized at spindle poles[51]. The anaphase-promoting complex/cyclosome (APC/C) activates separase and cleaves cohesin at centrosomes in mitotic exit, which in turn triggers the centriole disengagement. Importantly, the APC/C inhibitor EMI1 (early mitotic inhibitor) locally inhibits APC/C activity at spindle poles during mitosis and the spindle pole-localized EMI1 depends on the direct interaction with NuMA[52,53]. Considering these preceding studies, the two major defects, premature centriole disengagement and NuMA mislocalization, seen in *Dynlrb2* KO metaphase I could have a direct causal relationship; i.e., the NuMA mislocalization leads to EMI1 mislocalization, which leads to premature activation of APC/C at spindle poles and thus leads to premature centriole disengagement.

Why do male meiotic cells employ the male-germ cell-specific DYNLRB2 instead of mitotic DYNLRB1? Compared to mitotic spindles, which only need to capture and segregate sister chromatids, meiotic spindles in metaphase I need to capture and segregate cargo that is twice as large, the bivalent chromosomes[54]. Oocytes from most animal species undergo acentrosomal spindle assembly and form unique barrel-shaped spindles, probably to carry out this meiosis-specific chromosome segregation[55,56]. In male meiosis, spindle formation relies on the canonical centriole/centrosome system as seen in mitotic cells but might require meiosis-specific modifications for its meiosis-specific tasks. The robust DYNLRB2-dependent accumulation of NuMA to the spindle poles could be one such mechanism. In mitosis, the centrosome pathway and dynein–NuMA pathway redundantly promote spindle bipolarization, and thus NuMA depletion causes deleterious spindle defects, especially in acentrosomal mitotic cells[57].

It is possible that the MT's nucleation activity in male meiotic centrosomes could be weaker than that in mitotic cells, and this would mean that spindle bipolarity in spermatocytes largely depends on the dynein–NuMA pathway, which is ensured by testis-specific DYNLRB2.

Our analysis showed that the role of DYNLRB1 in mitosis is complemented by ectopically expressed DYNLRB2. However, it remains an open question whether DYNLRB1 can complement the role of DYNLRB2 in male meiotic cells. Our analysis implies that DYNLRB2 acts as a stronger localizer of NuMA compared to DYNLRB1 and therefore probably has a unique role in meiosis that is not fully replaceable by DYNLRB1. In future studies, the generation and analysis of genetically engineered mice expressing *Dynlrb1* instead of *Dynlrb2* in male-germ cells will provide further insights into the functional differences between DYNLRB1 and DYNLRB2 and the unique features in male meiotic spindle regulation.

## Methods

### Mice

*Dynlrb2*$^{-/-}$ mice (C57BL/6N-A < tm1Brd > Dynlrb2 < tm1a(KOMP)Wtsi > /WtsiOrl) were generated in this study. All WT and KO mice were congenic with the C57BL/6N background. The mice are housed in IVC cages with a 12 h dark and light cycle. The temperature is 20–22 °C and the relative humidity is between 45% and 60 %. The mice have bedding material in the form of wood shavings and wood litter as well as a house of paper and nesting pads as enrichment. Cage changing is done at least once a week. All animal experiments were approved by the Regional Ethics Committee of Gothenburg, governed by the Swedish Board of Agriculture (#1316/18).

### Histological analysis

Testes, epididymis, and brains were fixed in Bouin's fixative for 24 h at room temperature and embedded in paraffin blocks. Slices of 5–8 µm thickness were stained with hematoxylin and eosin. TUNEL analysis was carried out with an ApopTag Plus In Situ Apoptosis Fluorescein Detection Kit (S 7111; Millipore).

## Antibodies

The following antibodies were used: rabbit antibodies against DYNLRB2 (1:100 for IF; 1:1000 for WB; this study), DYNC1H1 (1:100 for IF; 1:1000 for WB; Proteintech; 12345-1-AP, 00080529), DYNC1I2 (1:100 for IF; BETHYL; A304-529A-T, 1), STIL (1:500 for IF; BETHYL; A302-442A, 1), DYNLL1 (1:100 for IF; 1:1000 for WB; Abcam; ab51603, GR3251355-4), KIF11 (Eg5) (1:100 for IF; Sigma; HPA010568, A118067), α-tubulin (1:1000 for IF; Abcam; ab18251, GR3406015-1), SYCP1 (1:1000 for IF; Abcam; ab15090, GR3184119-1), NuMA (1:100 for IF; Abcam; ab36999, GR296250-1), NuMA (1:100 for IF; 1:1000 for WB; Abcam; ab109262, GR154119-10), γ-tubulin (1:500 for IF; Abcam; ab11317, GR3415347-1), pericentrin (PCNT) (1:300 for IF; Abcam; ab4448, GR3354375-2), and NDC80 (1:100 for IF; this study); mouse antibodies against MLH1 (1:100 for IF; BD Biosciences; 51-1327GR, 4136717), β-ACTIN (1:1000 for IF; Sigma; A2228-200UL, 067M4856V), α-tubulin (1:1000 for IF; Abcam; ab7291, GR3398636-5), γ-tubulin (1:500 for IF; Abcam; ab27074, GR3246908-22), PLK1 (1:500 for IF; Abcam; ab17056, GR3260806-7), CENP-E (1:100 for IF; this study), DYCN1I1/2 (1:1000 for IF; Millipore; MAB1618, 3601722), SAS-6 (1:250 for IF; Santa Cruz; Sc-81431, C3021), and DYNC1LI2 (1:100 for IF; 1:300 for WB; this study); goat antibodies against DCTN1 (P150) (1:100 for IF; Abcam; ab11806, GR3359155-3); sheep antibodies against BubR1 (1:100 for IF; Abcam; ab28193, GR3205690-18); rat antibodies against CENTRIN2 (CETN2) (1:300 for IF; BioLegend; 698602, B333787); and chicken antibodies against SYCP3 (1:5000 for IF; Hiroki Shibuya lab).

## Antibody production

cDNAs encoding *Dynlrb2* (full length), *Ndc80* (full length), *Dycn1li2* (amino acids 246–492), and *Cenp-E* (amino acids 2381–2471) were cloned into the pET28c+ vector (Millipore). The HIS-tagged recombinant proteins were expressed in BL21 (DE3) cells, solubilized in a buffer of 600 mM NaCl, 30 mM imidazole, 20 mM Tris–HCl (pH 7.5), and 0.1% Triton X-100, and purified with Ni-NTA resin (Qiagen). The recombinant proteins were dialyzed in PBS and used to immunize the animals. The polyclonal antibodies were affinity purified on antigen-coupled Sepharose beads (GE Healthcare).

## Yeast two-hybrid assay

Yeast two-hybrid screening was performed by Hybrigenics Services, Paris, France. The coding sequence for *Dynlrb2* was cloned into pB27, and the construct was used as bait to screen a random-primed mouse testis cDNA library constructed in pP6. Using a mating approach with YHGX13 and L40ΔGal4 yeast strains, 70 million clones (6.5-fold the complexity of the library) were screened. Positive colonies were selected on a medium lacking tryptophan, leucine, and histidine and supplemented with 10 mM 3-aminotriazole. The prey fragments of the positive clones were amplified by PCR and sequenced. The resulting sequences were used to identify the corresponding interacting proteins in the GenBank database (NCBI) using a fully automated procedure. For the yeast two-hybrid assay, *Dynlrb1* and *Dynlrb2* cDNAs were cloned into the pGBKT7 vector. *Dync1i1* cDNAs were cloned into the pGADT7 vector. These baits and prey were co-transformed into the yeast strain AH109, and the positive transformants were selected on nutrition-restricted plates (SD-tryptophan-leucine-histidine-adenine).

## Reverse transcription PCR

Total RNA was isolated from tissues using the RNeasy Mini kit (Qiagen). cDNAs were generated by iScript reverse transcription super mix (Bio-Rad), and PCR amplification was performed using standard DNA polymerase. The primer information is provided in Supplementary Data 1.

## Immunostaining of spermatocytes

Testis cell suspensions were prepared by mincing the tissue with flat-head forceps in PBS, washing several times in PBS, and resuspending in a hypotonic buffer (30 mM Tris (pH 7.5), 17 mM trisodium citrate, 5 mM EDTA, 2.5 mM DTT, 0.5 mM PMSF, and 50 mM sucrose). After 30 min, the sample was centrifuged and the supernatant was aspirated. The pellet was resuspended in 100 mM sucrose. After 10 min, an equal volume of fixation buffer (1% paraformaldehyde and 0.1% Triton X-100) was added. Cells were applied to a glass slide, allowed to fix for 2 h at room temperature, and air-dried. For the analysis of spindle structures and NuMA localization in metaphase I spermatocytes, samples were prepared using a seminiferous tubule squash technique. In short, seminiferous tubules were placed on a glass slide, minced in a drop of fixative (2% paraformaldehyde and 0.1% Triton X-100), and incubated for 10 min. The coverslip was put on the slide and the sample was snap-frozen on dry ice. For immunostaining, the slides were incubated with primary antibodies in PBS containing 5% BSA for 2 h and then with the following secondary antibodies for 1 h at room temperature: Donkey Anti-Rabbit Alexa 488 (1:1000; Invitrogen; A21206, 2376850), Donkey Anti-Rabbit Alexa 594 (1:1000; Invitrogen; A21207,2313074), Donkey Anti-Mouse Alexa 594 (1:1000; Invitrogen; A21203, 2352146), Donkey Anti-Mouse Alexa 488 (1:1000; Invitrogen; A21202, 2309139), Goat Anti-Chicken Alexa 647 (1:1000; Invitrogen; A21449,1806124), Donkey Anti-Rat Alexa 594 (1:1000; Invitrogen; A21209, 1807726), Donkey Anti-Rat Alexa 488 (1:1000; Invitrogen; A21208, 1810450), Donkey Anti-Goat Alexa 488 (1:1000; Invitrogen; A32814, UE286661), and Donkey Anti-Sheep Alexa 594 (1:1000; Invitrogen; A11016, 1017334). The slides were washed with PBS and mounted in Vectashield medium with DAPI (Vector Laboratories).

## Preparation of testis extract and IP

Testes were removed from male C57BL/6J mice and suspended in extraction buffer (20 mM Tris–HCl (pH 7.5), 50 mM KCl, 0.4 mM EDTA, 5 mM MgCl$_2$, 10% glycerol, 0.1% Triton X-100, and 1 mM β-mercaptoethanol) supplemented with cOmplete Protease Inhibitor (Roche) and Phosphatase Inhibitor (Roche). After homogenization, the cell extract was centrifuged at 50,000 × $g$ for 30 min at 4 °C and the supernatant (chromatin extract) was isolated. The extract was supplemented with Dynabeads protein A (Thermo Fisher Scientific) conjugated with antibodies or control IgG and incubated for 3 h at 4 °C. The beads were washed with high-salt buffer (20 mM HEPES (pH 7.0), 400 mM KCl, 5 mM MgCl$_2$, 10% glycerol, 0.1% Triton X-100, and 1 mM β-mercaptoethanol) supplemented with cOmplete Protease Inhibitor (Roche) and Phosphatase Inhibitor (Roche). The samples were eluted with 0.1 M glycine (pH 2.5).

## Sucrose density gradient

Mouse testes were dissected, separated from the tunica, and homogenized in 210 μl of lysis buffer (20 mM Tris–HCl (pH 7.5), 50 mM KCl, 0.4 mM EDTA, 5 mM MgCl$_2$, 10% glycerol, 0.1% Triton X-100, and 1 mM β-mercaptoethanol) supplemented with cOmplete Protease Inhibitor (Roche). The sample was mechanically disrupted and homogenized in lysis buffer with gentle teasing apart and pipetting. The lysate was centrifuged at 13,000 × $g$ at 4 °C for 15 min. The supernatant was then layered onto continuous gradients containing 10% to 40% (w/v) sucrose prepared in a buffer containing 90 mM KCl, 50 mM HEPES/KOH (pH 7.2), and protease inhibitors. Gradients were centrifuged at 85,000 × $g$ for 18 h at 4 °C in a Beckman SW55 Ti rotor. Fourteen equal-volume fractions were collected and analyzed by western blotting.

## Cell culture

B16-F1 and HEK293 cell lines were maintained in DMEM (GIBCO Life Technologies) supplemented with 10% FBS (Invitrogen), 100 U/ml penicillin–streptomycin (GIBCO Life Technologies), and 2.5 μg/ml Plasmocin (InvivoGen) in a humidified atmosphere of 5% CO$_2$ at 37 °C. Mouse embryonic fibroblasts were isolated from embryonic day 13.5 embryos and cultured in DME containing 10% bovine calf serum at 37 °C and 5% CO$_2$. Ear fibroblast cells were obtained from adult mice

and cultured in DMEM with 10% FBS, penicillin–streptomycin, and Plasmocin. Cells were fixed in methanol and stained for further analysis. For RNAi knockdown experiments, the B16-F1 and HEK293 cell lines were grown for 24 h before treatment with either control siRNA or *DYNLRB1* siRNA. SiRNAs against different genes were synthesized by Thermo Fisher. MISSION siRNA Universal Negative Control #1 (Thermo Fisher) was used as the non-targeting siRNA control. The sequence information is provided in Supplementary Data 1. The siRNA transfection was performed using Lipofectamine 2000 transfection reagent (Invitrogen) and Optimem (GIBCO Life Technologies). The culture medium was then replaced by a fresh medium and incubated for 48 h. The mRNA expression levels were assessed by reverse transcription-quantitative polymerase chain reaction after transfection. For the rescue experiments, siRNAs were co-transfected with RNAi-resistant *Dynlrb1*, *hDYNLRB1*, *Dynlrb2*, or hDYNLRB2-expressing plasmids (1 μg/well) into cells. The control group was transfected with the control vector.

### Single-cell RNA sequencing transcriptome analysis

Single-cell RNA sequencing data of young mouse testes were obtained from previously published reports (E-code E-MTAB-6946)[26]. Sequence reads from PD5–25 WT mouse testis was aligned to the reference mouse genome data (mm10) using the 10x Genomics Cell Ranger count pipeline version 6.0.2 with the default settings[58], and the multiplexed samples were aggregated using the Cell Ranger aggr pipeline with default settings. Cell populations were clustered into seven groups by the *k*-means clustering method and plotted by uniform manifold approximation and projection (UMAP) using 10x Genomics Loupe Browser software version 6.2. Cell types or spermatogenic developmental stages of clustered populations were identified based on the expression pattern of known stage-specific marker genes, including *Stra8* for preleptotene spermatocytes, *Sycp3*, *Piwil1*, *Spo11*, and *Dmc1* for meiotic spermatocytes, *Rec8* for spermatocytes and round spermatids, *Sox9* and *Wt1* for Sertoli cells, and *Cyp11a1*, *Hsd3b1*, and *Insl3* for Leydig cells.

### Single-molecule imaging

HeLa cells stably expressing GFP-labeled dynein HC[59] were transfected with 2 μg ^FLAG-h*DYNLRB1* or ^FLAG-h*DYNLRB2* with Lipofectamine 2000 (Invitrogen) according to the manufacturer's instructions. After 48 h, the cells were washed with PBS and lysed in lysis buffer (30 mM HEPES (pH 7.4), 50 mM potassium acetate, 2 mM magnesium acetate, 1 mM EGTA, 10% glycerol, 1 mM DTT, 0.5 mM Mg-ATP, and 1× protease inhibitor cocktail (Roche)). Cells were incubated at 4 °C for 20 min before being extruded 5 times through a 20 G syringe and 8 times through a 25 G syringe (with incubations on ice interspersed throughout the extrusions). Lysates were centrifuged at 4 °C for 15 min at 21,000 × *g*. The supernatant was incubated with 50 μl of anti-FLAG M2 Affinity Gel (Millipore) at 4 °C for 90 min with slow rotation. The resin was washed extensively with lysis buffer, and the IP was eluted with lysis buffer supplemented with 0.4 mg/ml 3× FLAG peptide (ApexBio). The resulting elution was snap frozen in liquid nitrogen in small aliquots. Single-molecule imaging was performed with an inverted Ti2-E Eclipse microscope (Nikon) with a 100× 1.49 N.A. oil immersion objective (Nikon, Apo). The microscope had a LUNF-XL laser launch (Nikon) with 405, 488, 561, and 640 lasers. The excitation path was filtered with a quad bandpass filter cube (Chroma), and the emission path was filtered through appropriate filters (Chroma) housed in a high-speed filter wheel (Finger Lakes Instrumentation). Emission was detected on an iXon Ultra 897 electron-multiplying CCD camera (Andor Technology). NIS Elements Advanced Research software (Nikon) was used to control the microscope for image acquisition. Single-molecule motility assays were performed in flow chambers as described previously[60]. No. 1-1/2 coverslips (Corning) were sonicated in 100% ethanol for 10 min and dried before use. Taxol-stabilized

microtubules with ~10% biotin-tubulin and ~10% Alexa 647 conjugated-tubulin (Cytoskeleton) were prepared as described previously[61]. Flow chambers were assembled with Taxol-stabilized microtubules by incubating the following sequence of solutions interspersed with two 20 μl washes with assay buffer (30 mM HEPES (pH 7.4), 2 mM magnesium acetate, 1 mM EGTA, 10% glycerol, and 1 mM DTT supplemented with 20 μM Taxol): (1) 1 mg/ml biotin-bovine serum albumin (BSA) in assay buffer (3 min incubation), (2) 0.5 mg/ml streptavidin in assay buffer (3 min incubation), and (3) freshly diluted Taxol-stabilized microtubules in assay buffer (3 min incubation). After the step where the microtubules were added, the two interspersed 20 μl washes were performed with assay buffer supplemented with 1 mg/ml casein and 20 μM Taxol. To image the h*DYNLRB1/2* IPs, the elutions were thawed, supplemented with purified dynactin and BicD2, and diluted such that the final buffer consisted of 30 mM HEPES (pH 7.4), 12.5 mM KOAc, 1 mM DTT, 2.75 mM Mg-ATP, 2 mM magnesium acetate, 1 mM EGTA, 10% glycerol, 0.0025% Triton X-100, 0.25× protease inhibitor cocktail (cOmplete, Roche), 15 μM Taxol, 0.75 mg/ml casein, 71.5 mM β-mercaptoethanol, 0.05 mg/ml glucose catalase, 1.2 mg/ml glucose oxidase, 0.4% glucose, 4.5 nM dynactin, and 45 nM BicD2. All samples were imaged every 300 ms for 3 min. Three biological replicates, defined as separate transfections and immunoprecipitates, were performed.

### Microscopy

Images were obtained on a microscope (Olympus IL-X71 Delta Vision; Applied Precision) equipped with 100× NA 1.40 and 60× NA 1.42 objectives, a camera (CoolSNAP HQ; Photometrics), and softWoRx 5.5.5 acquisition software (Delta Vision). Images in Fig. 3e were processed with the deconvolution algorithm in softWoRx 5.5.5. All acquired images were processed with Photoshop (Adobe).

### Statistics and reproducibility

The experiments were not randomized, so no statistical method was used to predetermine sample size, and the investigators were not blinded to allocation during the experiments or to outcome assessment. Each experiment and conclusion in the manuscript was based on results that were reproduced in at least three independent experiments and in at least three independent mice of each genotype. Sample sizes, statistical tests, and *p*-values are indicated in the text, figures, and figure legends.

### Reporting summary

Further information on research design is available in the Nature Portfolio Reporting Summary linked to this article.

## Data availability

The authors declare that the data supporting the findings of this study are available within the paper and its Supplementary Information. Single-cell RNA sequencing data of young mouse testes were obtained from previously published reports and are available at ArrayExpress. Source data are provided with this paper. All other data supporting the findings of this study are available from the corresponding author upon reasonable request. Source data are provided with this paper.

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

## Acknowledgements

We thank Atsuo Ogura (RIKEN) and Tomomi Kiyomitsu (OIST) for their valuable discussions. We thank Kristina Hedfalk, Per Sunnerhagen, Marc Pilon, and Peter Carlsson (University of Gothenburg) for their continuous support of our research activities. J.G. acknowledges funding from the Swedish Cancer Society (CAN2018/538 and CAN2021/1755). M.E.D. acknowledges funding from the NIH (R00-GM127757). H.S. acknowledges funding from the European Research Council (StG-801659), the Knut and Alice Wallenberg Foundation (KAW 2019.0180), and the Swedish Research Council (2018-03426).

## Author contributions

S.H. performed most of the mouse experiments and analyzed the data; J.P.G., J.L.Z., and M.E.D. performed the IP-TIRF experiments; C.M.C. under the supervision of J.G. performed the sucrose density gradient; Y.F. analyzed the published single-cell RNA sequencing data; I.Y. contributed to the NuMA quantification in spermatocytes; and H.S. conceptualized and supervised the project and wrote the manuscript.

## Funding

## Competing interests

The authors declare no competing interests.
