## [Peer Review File · Nature Communications]

Distinct dynein complexes defined by DYNLRB1 and DYNLRB2 regulate mitotic and male meiotic spindle bipolarityREVIEWER COMMENTS

Reviewer #1 (Remarks to the Author):

In this manuscript, He et al. examined tissue-specific expression of different dynein light chains and identified DYNLRB2 as a testis-specific dynein light chain. The authors further demonstrated that DYNLRB2 is part of the dynein complexes and localizes to spindle poles in spermatocytes. By knocking out *Dynlrb2*, the authors found that male KO mice are infertile and accumulate metaphase I-arrested spermatocyte. Further analysis of the spindles in *Dynlrb2* KO spermatocytes revealed spindle pole defects, which are associated with premature disengagement of centrioles and NuMA localization to spindle poles. Finally, the authors knocked down the ubiquitously expressed *Dynlrb1* in somatic cells and found that *Dynlrb1* KD cells show similar spindle pole defects (as in *Dynlrb2* KO spermatocytes), which can be rescued by the expression of either DYNLRB1 or DYNLRB2. Together, the authors proposed that two distinct dynein complexes containing DYNLRB1 and DYNLRB2 specifically regulate mitotic and meiotic spindle bipolarity, respectively.

This manuscript will not only be of interest to the mitosis field, but also to the male meiosis field considering the lack of mechanistic studies on male meiotic spindles. Overall, the experiments were well designed and executed. Nevertheless, there are a few concerns that should be addressed prior publication.

Major concerns:

1. In lines 148 and 149, the authors wrote "...significant accumulation of the checkpoint component BubR1 at kinetochores (Fig. 3f), and this accounted for the observed metaphase I arrest phenotype". While the former statement was supported by Fig. 3f, the later statement was not supported by any experimental evidence and was a speculation. To draw this conclusion, the authors will need to confirm whether meiosis I progression in *Dynlrb2*^{-/-} spermatocytes resumes after inhibition of spindle assembly checkpoint pathway, e.g. with reversine.
2. In lines 249-251, the authors wrote "..., likely by linking NuMA to dynein and promoting retrograde transport, and that DYNLRB2 ensures the NuMA-dependent bundling of MT minus-ends thus preventing spindle pole collapse". This speculation should be moved to the discussion section.
3. In the abstract (lines 21-23) and discussion (lines 322-324), the authors concluded that the spindle pole phenotypes in *Dynlrb2* KO spermatocytes were caused by premature centriole disengagement and loss of NuMA enrichment at the poles. The authors should note that these observations are associated with the phenotypes, but the causal relationships between them were not established, particularly for the latter one.

In *Dynlrb2* KO spermatocytes, 23% of spindles were multipolar with separated centrioles (Fig. 3g). Because these separated centrioles were all associated with PCM (Fig. 3h) and could act as individual MTOCs, premature centriole disengagement could well justify for the formation of these multipolar spindles.

However, for the remaining 48% of spindles with multiple poles and paired centrioles (Fig. 3g), this reviewer does not agree with the authors that these can be justified by the loss of NuMA enrichment at the poles for two reasons. First, partial depletion of NuMA did not result in multipolar spindle formation (Haren et al., 2009), whereas full depletion of NuMA resulted in completely defocused spindle (Hueschen et al., 2017), at least in other centrosomal spindle systems. Second, in Fig. 5j, the multipolar spindle in *Dynlrb2* KO spermatocytes appeared to stain NuMA similarly to or more intensely than that in the bipolar spindle in *Dynlrb2* KO spermatocytes.

To further support this claim, the authors should provide evidence that partial depletion of NuMA in spermatocytes results in multipolar spindles with paired centrioles. Otherwise, the loss of NuMA enrichment at the poles could well just be an independent consequence of perturbing NuMA-dynein interactions by *Dynlrb2* KO.

Alternatively, the authors should consider the possibility that dynein prevents PCM fragmentation independent of NuMA. Indeed, knockdown of LIC1 and LIC2 can similarly result in PCM fragmentation without affecting NuMA enrichment at the poles (Jones and Villemant et al., 2014).

Minor concerns:

1. In line 113, the authors wrote "Dynlrb2^{-/-} female mice were fertile, but Dynlrb2^{-/-} male mice were completely infertile". The authors should provide information on e.g. no. of mating tested and ideally statistics to support this conclusion.
2. In line 118, the authors wrote "Consequently, there were no spermatozoa in the Dynlrb2^{-/-} epididymis (Fig. 2d)" and showed a representative section featuring a single seminiferous tubules in Fig. 2d. The authors should provide quantification e.g. % of seminiferous tubules containing spermatozoa to support this conclusion.
3. In Fig. 3a, 3f and 3g, the multipolar spindles showed did not look like the classical multipolar spindles with fully separated poles (e.g. in 4b, 4e, 5j, 6d), but rather a more or less bipolar spindle with broader poles. Were these representative images or just an impression caused by orthogonal projection of z-stack?
4. In lines 151 and 152, the authors wrote "In the Dynlrb2^{-/-} metaphase I spindle, the astral microtubules were completely depleted (Fig. 3g)" and showed a representative image in Fig. 3g. The authors should at least show a line scan analysis or ideally provide quantification.
5. In Fig. 5i, the protein probed was not labelled for the fifth blot.
6. In Fig. 5j and 6h, the authors should specify which spindles were used for line scan analyses and use arrows to illustrate the orientation and direction of line scan analyses.
7. In line 269, the authors wrote "the fragmented PCM was always accompanied by CETN2 foci (Fig. 6g)". Because this was quite different from the case of Dynlrb2 KO spermatocytes, where some fragmented PCMs were accompanied by CETN2 foci and some were not, the authors should provide proper quantification e.g. % of PCM foci accompanied by CETN2 foci for Dynlrb1 KD cells.

Reviewer #2 (Remarks to the Author):

This study submitted by He et al., is focused on a meiosis-specific role for the Roadblock cytoplasmic dynein light chains in spindle assembly and function. An open question in the field is why so many variants of the dynein light chains exist. This study provides a compelling explanation for this diversity. The authors determine that Roadblock 1 (Rb1) and Roadblock 2 (Rb2) are expressed in a mutually-exclusive manner: Rb1 is expressed in mitotic cells, whereas Rb2 is expressed in male meiotic cells. In both cases, the RB dynein light chains appear to concentrate in intact dynein complexes. Mouse KO studies suggest that Rb2 expression is required for formation of focused and intact meiotic spindles, and that Rb2 loss blocks meiotic progression by driving defects in meiosis 1. Mechanistically, the authors show that recruitment of NuMa to meiotic spindle poles requires Rb2, and that the loss of NuMa concentration at these poles can explain the defects observed in Rb2 (-/-) mice. Rb1, on the other hand, serves a similar function in mitotic cells.

Overall, I found this study convincing and think that it provides an important advance for the field. The mutually-exclusive nature of Rb1 and Rb2 in mitosis vs. meiosis is interesting in the context of dynein subpopulation functions - something that has not been examined well by others. The conclusions are supported by the data presented, and the KO mouse work was a powerful addition. I found the use of illustrations to explain the imaging data helpful. Overall, I support publication of this study.

Minor concerns:

- 1) The authors do an admirable job putting their work in context with previous studies on NuMa and potential interactions with dynein. However, data that supports the relative contributions of Rb light chains vs. dynein light intermediate chains would strengthen their argument. Some quantitative data that compares Rb1 vs. Rb2 complexes and NuMa binding would serve this purpose.
- 2) Some form of western blot quantification would bolster conclusions.

Reviewer #3 (Remarks to the Author):

In this study, He et al. study the roles of DYNLRB1 and DYNLRB2, two dynein light chains, in the formation and integrity of the mitotic and male meiotic spindle respectively. By analyzing mRNA and protein expression in mice, they show that DYNLRB1 is ubiquitously expressed in somatic cells, while DYNLRB2 expression is restricted to the brain, lung and testis. They further find that within testis, DYNLRB2 is restricted to meiotic cells and is found in complex with other dynein chains, likely forming a functional dynein motor complex. Accordingly, DYNLRB2 was found at the meiotic spindle poles of metaphase I and II spermatocytes. By generating a mouse knock-out of DYNLRB2, the authors show that it is required for male fertility and that the infertility phenotype is caused by a depletion of round spermatids from the seminiferous tubules. Analysis of apoptotic cells further showed that DYNLRB2 is required for progression beyond the metaphase I stage. Immunofluorescent analyses of various spindle and centrosomal markers in DYNLRB2 KO spermatocytes demonstrated that spindle poles are disorganized, with frequent multipolar spindles, due to premature disengagement of mother and daughter centrioles. Furthermore, the authors suggest that this premature centriolar splitting is due to a defect in NuMA accumulation at spindle poles, due to its lack of dynein-mediated polar transport, which in turn would lead to centriole disengagement. Finally, the role of DYNLRB1 in mitotic somatic cells is studied by RNAi. A similar multipolar spindle phenotype is reported. However, unlike DYNLRB2 in meiosis, the DYNLRB1 phenotype is proposed to be caused by centriole overduplication, instead of premature disengagement.

Overall, this is an interesting study with an impressive amount of work, which demonstrates the differential requirement for two paralogous dynein light chains in mitosis and male meiosis. The study is however purely descriptive and the proposed model, although convincing, based on incremental conclusions rather than clear demonstrations. This however stems from the specific cell type studied (i.e. spermatocytes in metaphase), which are hardly amenable to live imaging or functional perturbations. Listed below are suggestions and minor points that would improve this manuscript:

- 1- The authors propose that the centriole disengagement phenotype observed in *dynlrb2*^{-/-} spermatocytes is due to EMI1 mislocalization, itself due to the lack of normal NuMA accumulation at spindle poles. This should be directly assessed by analyzing EMI1 localization in knock-out spermatocytes.
- 2- Many of the images of spindles presented as "bipolar" do not look bipolar to me (fig 4e, fig 5b-h). In panel 4e, for example, the image shown does not correspond to the schematic on the right. Furthermore chromosomes are spread all over the cell in all these pictures, which are supposed to represent Metaphase I spermatocytes. Could the authors clarify these points, as their main reported phenotype is a defect in spindle pole organization ?
- 3- What are the faint DYNLRB2 foci visible in Extended data fig 3d in the *Dynlrb2*^{-/-} panel ?
- 4- Based on the presence of more than four PCNT foci in *Dynlrb1* KD cells, the authors conclude that PCM fragmentation is caused by centriole overduplication, rather than premature disengagement. An alternative hypothesis could be a failure in cytokinesis of *Dynlrb1* KD cells. This is consistent with the tripolar midbody phenotype presented in fig 6f. This should be directly assessed by performing live imaging on *Dynlrb1* KD cells during mitosis.
- 5- Figure 6 should be better organized. Currently panel i is presented after j and k.
- 6- The second to last line is unlabeled in fig 5i.
- 7- A supplementary movie should be provided for the dynein mobility assay presented in fig 6l.
- 8- Line 324: edit "mislocalization".

We thank the editor and reviewers for providing favourable responses and constructive comments on our manuscript. Please find below our point-by-point responses to all the reviewer's comments.

Reviewer #1

Major concerns:

1. In lines 148 and 149, the authors wrote “...significant accumulation of the checkpoint component BubR1 at kinetochores (Fig. 3f), and this accounted for the observed metaphase I arrest phenotype”. While the former statement was supported by Fig. 3f, the later statement was not supported by any experimental evidence and was a speculation. To draw this conclusion, the authors will need to confirm whether meiosis I progression in *Dynlrb2*^{-/-} spermatocytes resumes after inhibition of spindle assembly checkpoint pathway, e.g. with reversine.

The normal development of mammalian spermatocytes requires the *in vivo* environment with the supporting somatic cells in testis tissue and, thus, there is no established method for the live-cell imaging of metaphase I and metaphase II progression in mammalian spermatocytes. Therefore, it is technically very difficult (if not impossible) to conduct meiotic cell synchronization, Reversine treatment, and live imaging to monitor the progression beyond metaphase I arrest in our case. This is a huge technical difference from many preceding studies using mitotic cell lines, which is also recognized by reviewer #3.

In addition to the technical barrier, different from mitotic cells, which can be arrested for a long time in M phase by the activation of spindle assembly checkpoint (SAC), the metaphase I arrested mouse spermatocytes in testis provoke apoptosis as seen by the accumulation of TUNEL-positive metaphase I spermatocytes in the *Dynlrb2* KO seminiferous tubules (see Fig. 2e, 2f, and below new data). Thus, metaphase progression less likely resumes after Reversine treatment in the case of metaphase I-arrested *Dynlrb2* KO spermatocytes in testis.

DAPI TUNEL

Due to the above two reasons, the staining and quantification of the BubR1-positive metaphase I cell population in the fixed cell samples, as we did in Fig.3f, has been widely used as the indicator of SAC activation and concomitant metaphase I arrest in mouse spermatocyte studies (for example: Fig.3 of PMID: 31533924).

Therefore, we are thinking that current dataset is sufficient to show the activation of the spindle assembly checkpoint (fig 3.f) and the concomitant metaphase I arrest (fig. 2g-h). However, we acknowledge the reviewer's comment and changed the sentence in order not to emphasize the direct causal relationship between the SAC activation and the observed metaphase I arrest for accuracy.

Before

"the cells activated the spindle assembly checkpoint as indicated by the significant accumulation of the checkpoint component BubR1 at kinetochores (Fig. 3f), and this accounted for the observed metaphase I arrest phenotype."

After

"the cells activated the spindle assembly checkpoint as indicated by the significant accumulation of the checkpoint component BubR1 at kinetochores (Fig. 3f)."

2. In lines 249-251, the authors wrote "..., likely by linking NuMA to dynein and promoting retrograde transport, and that DYNLRB2 ensures the NuMA-dependent bundling of MT minus-ends thus preventing spindle pole collapse". This speculation should be moved to the discussion section.

We have a similar discussion in the discussion section, thus, we simply deleted the sentence from the result section as pointed out by the reviewer.

Before

"Thus we conclude that DYNLRB2 is crucial for the proper recruitment of NuMA to meiotic spindle poles, likely by linking NuMA to dynein and promoting its retrograde transport, and that DYNLRB2 ensures the NuMA-dependent bundling of MT minus-ends thus preventing spindle pole collapse."

After

"Thus we conclude that DYNLRB2 is crucial for the proper recruitment of NuMA to meiotic spindle poles and for the maintenance of spindle pole integrity."

3. In the abstract (lines 21-23) and discussion (lines 322-324), the authors concluded that the spindle pole phenotypes in Dynlrb2 KO spermatocytes were caused by premature centriole disengagement and loss of NuMA enrichment at the poles. The authors should note that these observations are associated with the phenotypes, but the causal relationships between them were not established, particularly for the latter one.

In Dynlrb2 KO spermatocytes, 23% of spindles were multipolar with separated centrioles (Fig. 3g). Because these separated centrioles were all associated with PCM (Fig. 3h) and could act as individual MTOCs, premature centriole disengagement could well justify for the formation of these multipolar spindles.

However, for the remaining 48% of spindles with multiple poles and paired centrioles (Fig. 3g), this reviewer does not agree with the authors that these can be justified by the loss of NuMA enrichment at the poles for two reasons. First, partial depletion of NuMA did not result in multipolar spindle formation (Haren et al., 2009), whereas full depletion of NuMA resulted in completely defocused spindle (Hueschen et al., 2017), at least in other

centrosomal spindle systems. Second, in Fig. 5j, the multipolar spindle in *Dynlrb2* KO spermatocytes appeared to stain NuMA similarly to or more intensely than that in the bipolar spindle in *Dynlrb2* KO spermatocytes.

To further support this claim, the authors should provide evidence that partial depletion of NuMA in spermatocytes results in multipolar spindles with paired centrioles. Otherwise, the loss of NuMA enrichment at the poles could well just be an independent consequence of perturbing NuMA-dynein interactions by *Dynlrb2* KO.

Alternatively, the authors should consider the possibility that dynein prevents PCM fragmentation independent of NuMA. Indeed, knockdown of LIC1 and LIC2 can similarly result in PCM fragmentation without affecting NuMA enrichment at the poles (Jones and Villemant et al., 2014).

The degree of spindle pole defects after NuMA deletion varies between cell types as shown in many preceding studies. An excellent study from Dr. Kitagawa's lab published in 2020 (PMID: 31782546) explained this by showing that the centrosomal pathway and dynein-NuMA pathway redundantly promote spindle bipolarization and that the relative contribution of the two pathways varies between cell types resulting in the phenotypic diversity (e.g., the NuMA depletion causes deleterious spindle defects especially in acentrosomal mitotic cell).

Regarding mammalian male spermatocytes, the contribution of the NuMA pathway for the spindle pole bipolarization has not been addressed in any preceding studies. This is because most of the studies in the field of spindle regulations have focused on mitotic cells (or female oocytes) and there have been few studies focusing on male spermatocytes, and thus our current study is a pioneering study in this field. For example, as far as we know, the immunolocalization of NuMA in male metaphase I spindle pole has been investigated for the first time in our current paper (Fig.5j). Further, we have shown that this NuMA localization at spindle poles largely depends on DYNLRB2, and thus the depletion of DYNLRB2 leads to the mislocalization of NuMA and concomitant spindle pole collapse (Fig.5j).

However, as the reviewer pointed out, we agree with the idea that "the causal relationship" between the NuMA mislocalization and the spindle pole collapse was not fully proven. In Fig. 5, we immunostained plenty of candidate proteins, including kinetochore proteins (NDC80, CENP-E, and PLK1,) and spindle pole proteins (Eg5, DYNLL1, p150, DYNC1H1, and NuMA) and found that none of these proteins, except for NuMA, was affected in *Dynlrb2* KO spermatocytes compared to the WT spermatocytes. Thus we reasoned that the NuMA mislocalization is likely to be a cause of the observed spindle defects, and **this interpretation is reasonable considering the established role of NuMA for the bundling of microtubule minus ends at spindle poles** in mitotic cells.

As the reviewer pointed out, the analysis of NuMA or LIC1/2 deficiencies in male meiosis will further help to support the causal relationship. To this end, because there has been no established method for efficient knockdown experiments in live mouse testis, we need to make testis-specific conditional knockout (or partial knockout, like heterozygous) mice for mouse *Numa* and *Dync1li1/2* genes. We believe that making these genetically modified animals (which will take more than a year of additional works) to further support our current findings is beyond the scope of our current paper. It would, however, be interesting to make

these genetically modified animals to further explore the regulation of male meiotic spindles in future follow-up studies.

Minor concerns:

1. In line 113, the authors wrote “*Dynlrb2*^{-/-} female mice were fertile, but *Dynlrb2*^{-/-} male mice were completely infertile”. The authors should provide information on e.g. no. of mating tested and ideally statistics to support this conclusion.

We have added a new graph showing the quantification of the number of pups from WT male vs KO female crossings in Supplementary fig. 3e with Student’s t-test. The data clearly support our conclusion that the female KO mice are fertile.

Supplementary fig. 3e

Legend

(e) The average number of pups per litter. PD60 male (M) and female (F) mice with indicated genotypes were paired for more than 90 days of continuous breeding. n indicates the number of mating pairs examined. All analyses used two-tailed t-tests. ns: not significant.

Regarding the fertility assay for the KO males mice crossed with the WT females, we found that the mice were completely sterile after more than 90 days of continuous breeding using more than 3 pairs for mating. There were no pups at all so we did not include the data in the quantification in Supplementary fig. 3e. The complete sterility of KO males was convincingly supported by the complete absence of mature spermatozoon in the KO epididymis (Fig. 2d, and also see the below additional data).

2. In line 118, the authors wrote “Consequently, there were no spermatozoa in the *Dynlrb2*^{-/-} epididymis (Fig. 2d)” and showed a representative section featuring a single seminiferous tubules in Fig. 2d. The authors should provide quantification e.g. % of seminiferous tubules containing spermatozoa to support this conclusion.

We showed a single epididymal tubule section from both WT and *Dynlrb2* KO epididymis as representative pictures in Fig. 2d. As requested by the reviewer, we show pictures of lower magnification as below. These pictures clearly show that the defect is 100% penetrance and there are no mature spermatozoa in *Dynlrb2* KO epididymis. Further, as requested, we also performed the quantification. Spermatozoa were observed in 99.8% (n = 5,114) and 0% (n =

3,828) of epididymal tubule sections in WT and *Dynlrb2* KO epididymides, respectively. We have added the quantification results in the figure legend.

	WT			Dynlrb2 ^{-/-}		
	Replicate 1	Replicate 2	Replicate 3	Replicate 1	Replicate 2	Replicate 3
Tubule containing spermatozoa (%)	99.6	99.9	100	0	0	0
Tubule number	1322	2026	1766	1025	1258	1545

Figure 2d legend

“Spermatozoa were observed in 99.8% (n = 5,114 tubules) and 0% (n = 3,828 tubules) of epididymal tubule sections in WT and *Dynlrb2*^{-/-} epididymides, respectively.”

3. In Fig. 3a, 3f and 3g, the multipolar spindles showed did not look like the classical multipolar spindles with fully separated poles (e.g. in 4b, 4e, 5j, 6d), but rather a more or less bipolar spindle with broader poles. Were these representative images or just an impression caused by orthogonal projection of z-stack?

Thank you for making this point clear. The different visual impressions of multipolar spindles in our pictures are indeed due to the orthogonal projection of z-stacks (please also find our response to Reviewer #3). Especially, the z-stack images are largely different in their visual impression between figures because we are using two different sample preparation methods depending on the purposes. Please find the below schematics showing the two methods.

Chromosome spreading

- Commonly used for mammalian spermatocyte immunostainings
- Better immunostaining outcomes in most of the antibody cases
- Partially impaired 3D structures

Squash technique

- Used exceptionally for some specific cases
- Largely preserved 3D structure of the cells
- Worse immunostaining outcomes in most of the case
- In some rare case, some antibody (such as NuMA in this study) works better than the spreading

Conventionally, the chromosome spreading method has been widely used in mammalian spermatocyte studies (for example: PMID: 29286440). This is because samples prepared by the chromosome spreading method work better for the immunostaining and result in better resolution in most of the cases. We are using this method for most of the immunostaining experiments in our paper such as (Fig. 4b, 4e).

However, to better preserve the 3D structure of the spindles, we sometimes used another method called the squash technique. In this method, spermatocytes were stained within the seminiferous tubules and thus their 3D structures were better preserved. At the cost of the better 3D preservation, the antibody accessibility to the cells became worse and the immunostaining outcomes were not so good in most of the cases. We are using this method for some exceptional purposes such as to observe the 3D spindle structures (fig. 3a, 3f, and 3g) or to stain NuMA (we found that the NuMA antibody worked much better in the squash technique compared to the spreading method).

We have already clarified the usage of the two distinct sample preparation methods in the material and method section. Moreover, in every figure legend where we used the squash technique, we have clarified the point as below.

For example;

Fig. 3 (a) Immunostaining of metaphase I spermatocytes **prepared by the squash technique**.

4. In lines 151 and 152, the authors wrote “In the *Dynlrb2*^{-/-} metaphase I spindle, the astral microtubules were completely depleted (Fig. 3g)” and showed a representative image in Fig. 3g. The authors should at least show a line scan analysis or ideally provide quantification.

Thank you for pointing this out. We performed the quantification of signal intensity (astral MTs vs polar MTs) and showed the significant reduction of the astral MTs in *Dynlrb2*^{-/-} metaphase I spindles. We have updated Fig. 3g.

Before

Fig. 3g

After

Fig. 3g

(g) Immunostaining of metaphase I spermatocytes prepared by the squash technique. Scale bar: 5 μ m (1 μ m, magnified panel). The graph shows quantification of astral MT intensity normalized by the polar MT intensity. All values are normalized by the average value of the WT controls. The mean values with SD are shown. Data were pooled from three independent stainings using three different mice for each genotype (n = 64 spindle poles from 32 cells for both WT and *Dynlr2*^{-/-}). Only cells that formed bipolar spindles were quantified. The analysis used two-tailed *t*-tests. *****p* < 0.0001.

5. In Fig. 5i, the protein probed was not labelled for the fifth blot. Thank you for pointing this out. We have fixed the mistake.

Before

After

6. In Fig. 5j and 6h, the authors should specify which spindles were used for line scan analyses and use arrows to illustrate the orientation and direction of line scan analyses. Thank you for pointing this out. We have clarified these points by changing the figures as below.

Before

Fig. 5j

Fig. 5i

After

Fig. 5j

Fig. 5i

Before

Fig. 6h

After

Fig. 6i

7. In line 269, the authors wrote “the fragmented PCM was always accompanied by CETN2 foci (Fig. 6g)”. Because this was quite different from the case of *Dynlrb2* KO spermatocytes, where some fragmented PCMs were accompanied by CETN2 foci and some were not, the authors should provide proper quantification e.g. % of PCM foci accompanied by CETN2 foci for *Dynlrb1* KD cells.

We have repeated the KD experiments another three times and quantified the % of PCNT foci accompanying CETN2 foci as below. Indeed, in *Dynlrb1* KD multipolar cells (with fragmented PCNT), almost all PCNT foci accompanied CETN2 foci, suggesting that the presence of supernumerary centrioles is the major reason for the observed multipolarity. We have shown multiple representative pictures below and added the quantification to the main figure as Fig. 6h. We have also modified the corresponding text as below.

Before

“Different from the *Dynlrb2*^{-/-} meiotic cells, all *Dynlrb1* KD cells with fragmented PCM had more than four CETN2 foci, and the fragmented PCM was always accompanied by CETN2 foci (Fig. 6g), suggesting that the PCM fragmentation was primarily caused by the overduplication of centrioles in *Dynlrb1* KD cells.”

After

“Different from the *Dynlrb2*^{-/-} meiotic cells, *Dynlrb1* KD cells with fragmented PCM had more than four CETN2 foci (Fig. 6g), and the quantification showed that the fragmented PCM was mostly accompanied by CETN2 foci (Fig. 6h), suggesting that the PCM fragmentation was primarily caused by the overduplication of centrioles in *Dynlrb1* KD cells.”

Reviewer #2 (Remarks to the Author):

Minor concerns:

- 1) The authors do an admirable job putting their work in context with previous studies on NuMa and potential interactions with dynein. However, data that supports the relative contributions of Rb light chains vs. dynein light intermediate chains would strengthen their argument. Some quantitative data that compares Rb1 vs. Rb2 complexes and NuMa binding would serve this purpose.
- 2) Some form of western blot quantification would bolster conclusions.

Thank you for your point. The formation of multipolar spindles in the mitotic cells after the treatment with siRNAs against human dynein light intermediate chains (DYNC1LI1/2) was already reported in a preceding study (PMID: 25422374). Furthermore, another comprehensive research paper already reported the systematic comparison of knockdown phenotypes for distinct dynein subunits in human mitotic cells and found that the knockdown of heavy chain (DHC), intermediate chain (DIC2), and DYNLRB1 showed spindle focusing defects to a comparable degree (30–40% in all cases, Fig. 7 of PMID: 23589491).

However, due to the differences in the knockdown efficiencies between different siRNAs, we need to be cautious to interpret this kind of quantitative data. Ideally, the generation of knockout cell lines by CRISPR-Cas9 should be preferential for this purpose to avoid the misinterpretation caused by differences in the knockdown efficiency.

However, I would like to argue that the focus of our current paper is the discovery of mutually exclusive usage of the two light chain paralogs, DYNLRB1 and DYNLRB2, in mitotic and meiotic cells, respectively, but not the distinct contribution of individual dynein subunits (such as DYNLRB1 vs light intermediate chain) in mitotic spindle formation. Thus, we think the experiments are beyond the scope of our current paper.

Regarding the quantitative comparison between DYNLRB1 and DYNLRB2, we performed an additional experiment. We transfected GFP-DYNLRB1 and GFP-DYNLRB2 into B16-F1 cells and performed GFP immunoprecipitations. We found that both GFP-DYNLRB1 and GFP-DYNLRB2 successfully co-immunoprecipitated other dynein subunits as well as NuMA suggesting that they formed dynein-NuMA complexes in a comparable manner.

We further extended the analysis by performing the quantitative mass-spectrometry analysis (see below additional data). Consistent with the Western blot results, we detected other dynein subunits and NuMA comparably in the two IPs. Interestingly, several actin-binding proteins (shown in yellow in the below graph) were specifically enriched in GFP-DYNLRB2 IP. These DYNLRB2-specific interactors could be important for the unique role of DYNLRB2 in male meiosis. However, at this point we have no clear explanations for this, and thus to avoid misinterpretations we would like not to show this data in the current paper. Future follow up studies focusing on the unique interactors for DYNLRB1 and DYNLRB2 could provide the unique roles of each dynein subcomplex.

Reviewer #3 (Remarks to the Author):

In this study, He et al. study the roles of DYNLRB1 and DYNLRB2, two dynein light chains, in the formation and integrity of the mitotic and male meiotic spindle respectively. By analyzing mRNA and protein expression in mice, they show that DYNLRB1 is ubiquitously expressed in somatic cells, while DYNLRB2 expression is restricted to the brain, lung and testis. They further find that within testis, DYNLRB2 is restricted to meiotic cells and is found in complex with other dynein chains, likely forming a functional dynein motor complex. Accordingly, DYNLRB2 was found at the meiotic spindle poles of metaphase I and II spermatocytes. By generating a mouse knock-out of DYNLRB2, the authors show that it is required for male fertility and that the infertility phenotype is caused by a depletion of round spermatids from the seminiferous tubules. Analysis of apoptotic cells further showed that DYNLRB2 is required for progression beyond the metaphase I stage. Immunofluorescent analyses of various spindle and centrosomal markers in DYNLRB2 KO spermatocytes

demonstrated that spindle poles are disorganized, with frequent multipolar spindles, due to premature disengagement of mother and daughter centrioles. Furthermore, the authors suggest that this premature centriolar splitting is due to a defect in NuMA accumulation at spindle poles, due to its lack of dynein-mediated polar transport, which in turn would lead to centriole disengagement. Finally, the role of DYNLRB1 in mitotic somatic cells is studied by RNAi. A similar multipolar spindle phenotype is reported. However, unlike DYNLRB2 in meiosis, the DYNLRB1 phenotype is proposed to be caused by centriole overduplication, instead of premature disengagement.

Overall, this is an interesting study with an impressive amount of work, which demonstrates the differential requirement for two paralogous dynein light chains in mitosis and male meiosis. The study is however purely descriptive and the proposed model, although convincing, based on incremental conclusions rather than clear demonstrations. This however stems from the specific cell type studied (i.e. spermatocytes in metaphase), which are hardly amenable to live imaging or functional perturbations. Listed below are suggestions and minor points that would improve this manuscript:

1- The authors propose that the centriole disengagement phenotype observed in *dynlrb2*^{-/-} spermatocytes is due to EMI1 mislocalization, itself due to the lack of normal NuMA accumulation at spindle poles. This should be directly assessed by analyzing EMI1 localization in knock-out spermatocytes.

Thank you for your point. The mislocalization of EMI1 followed by NuMA mislocalization can be a cause of the premature centriole disengagement seen in the *Dynlrb2* KO spermatocytes. This hypothesis can provide the direct causal relationship between the two major defects (the NuMA mislocalization and premature disengagement) seen in the *Dynlrb2* KO spermatocytes. We have mentioned this hypothesis only in the discussion section but not in the results section because we do not have enough data to support this hypothesis, and thus the point is not the main argument of our current paper.

As the reviewer suggested, the immunostaining of EMI1 could be one of the experiments to test this hypothesis. The problem is that all preceding studies focusing on EMI1 in mitotic spindle context used human cell lines and all the working antibodies used in these papers are against the human EMI1 protein but not the mouse EMI1 protein. To study the mouse spermatocytes, antibodies against mouse EMI1 that work for immunostaining are necessary.

This has been a common technical challenge throughout the current manuscript because most of the antibodies (for dynein subunits, kinetochore proteins, and centrosomal proteins) used in the preceding mitotic studies are against human proteins. Therefore, it has been a huge effort to identify antibodies that work for murine samples. We have used as many as 21 commercial antibodies in this paper as seen in the material method section (but we bought roughly twice more and found the other ones did not cross-react with mouse proteins). In addition to buying and testing a number of commercial antibodies, we have generated 4 polyclonal antibodies (against mouse DYNLRB2, NDC80, DYCN1LI2, and CENP-E) in this study to overcome the challenge.

Regarding the commercial EMI1 antibody, we bought one of the most promising commercial antibodies from Santa Cruz (sc-365212) for this revision purpose. This antibody is against

human EMI1 protein, but the antigen region shared around 70% sequence identity with the mouse EMI1 homolog. Even though this seems to be the most promising antibody for the detection of mouse EMI1 among the commercially available antibodies as far as we searched, we cannot detect any specific signals on meiotic spindles or spindle poles either in WT or *Dynlrb2* KO metaphase I spermatocytes (see below pictures) after the immunostainings, suggesting that this antibody is not working for detecting mouse EMI1 in the immunostaining analysis.

Considering the lack of working antibodies against the mouse EMI1 protein, we think that testing this hypothesis by newly generating antibodies (which will take 3–5 months, from antigen purification to immunization) is beyond the scope of our current paper. Accordingly, we have mentioned this hypothesis only in the discussion section but not in the results section.

2- Many of the images of spindles presented as “bipolar” do not look bipolar to me (fig 4e, fig 5b-h). In panel 4e, for example, the image shown does not correspond to the schematic on the right. Furthermore chromosomes are spread all over the cell in all these pictures, which are supposed to represent Metaphase I spermatocytes. Could the authors clarify these points, as their main reported phenotype is a defect in spindle pole organization ?

Thank you for making this point clear. The same issue is pointed out by reviewer #1. The different visual impressions of spindles in our pictures are due to the orthogonal projection of z-stacks. Especially, the z-stack images are largely different in their impression between pictures because we are using two different sample preparation methods depending on the purpose. Please find below the schematics showing the two methods.

Chromosome spreading

- Commonly used for mammalian spermatocyte immunostainings
- Better immunostaining outcomes in most of the antibody cases
- Partially impaired 3D structures

Squash technique

- Used exceptionally for some specific cases
- Largely preserved 3D structure of the cells
- Worse immunostaining outcomes in most of the case
- In some rare case, some antibody (such as NuMA in this study) works better than the spreading

Regarding the Fig. 4e, in order to immuno-stain centriolar proteins, we needed to apply the chromosome spreading rather than squash technique (because chromosome spreading yields better staining outcomes in most cases, including our centrin antibody case). Bipolar spindles prepared by the spreading method typically look like Fig. 4e, and this is because spindles are frequently compressed orthogonally to the slide glass during the spreading procedures as shown in the schematics below.

Chromosome spreading

3- What are the faint DYNLRB2 foci visible in Extended data fig 3d in the *Dynlrb2*^{-/-} panel ?

In Extended data Fig 3d (Supplementary Fig. 3d in the revised version), we showed that the signals at spindle poles and kinetochores stained by our DYNLRB2 antibody disappeared in the *Dynlrb2* KO spermatocytes, suggesting that these signals are genuine DYNLRB2 signals. As pointed out by the reviewer, we can still see the faint ignorable level of signals remaining in some of the *Dynlrb2* KO cells. The DYNLRB2 protein expression was completely abolished in the KO testis (Fig. 2b), thus, the remaining faint signals in the *Dynlrb2* KO spermatocytes are most likely background signals derived from the non-specific bindings of the antibodies to the cells. Normally, people could cut off this kind of faint background signal by applying cut-off threshold equally to WT and KO pictures. However, we did not want to apply such a strong contrast to show more natural pictures reflecting the real situation (it is very natural and often the case that the polyclonal antibodies yield some background signals). Another possibility is that our antibody against DYNLRB2 can recognize the close paralog DYNLRB1, and the residual signals could be derived from the cross-reaction with DYNLRB1. To rule out this possibility, we have purified recombinant his-tagged DYNLRB1 and DYNLRB2 (below left picture) and performed the Western blotting (below right picture). As you can

see in the pictures, our self-made DYNLRB2 antibody used in this study was highly specific to DYNLRB2 and there was no cross-reaction with DYNLRB1. As a control, we also tested a commercial antibody against human/mouse DYNLRB1 (Invitrogen, PA5-90288), but this antibody was not specific at all and recognized both DYNLRB1/2. In conclusion, our DYNLRB2 antibody used in this paper is highly specific, and the residual faint signals in *Dynlrb2* KO spermatocytes are just a non-specific background generally seen in many immunostainings using polyclonal antibodies.

4- Based on the presence of more than four PCNT foci in *Dynlrb1* KD cells, the authors conclude that PCM fragmentation is caused by centriole overduplication, rather than premature disengagement. An alternative hypothesis could be a failure in cytokinesis of *Dynlrb1* KD cells. This is consistent with the tripolar midbody phenotype presented in fig 6f. This should be directly assessed by performing live imaging on *Dynlrb1* KD cells during mitosis.

We fully agree with the idea that the observed cells with supernumerary centrioles (more than 4 Centrin foci) can be a consequence of cytokinesis errors. The same phenotype and the interpretation were already reported in a preceding study (Fig. 7 of PMID: 25422374). In this paper (PMID: 25422374), the authors knocked-down human dynein light intermediate chains (DYNC1L1/2) in mitotic cells and found abnormal cells with the supernumerary centrioles. The authors interpreted this by saying that “The small increase in cells with more than four centrioles likely reflects cells that had previously failed cytokinesis.”

In this sense, the appearance of cells with supernumerary centrioles in our *Dynlrb1* KD cells is a common phenotype seen in mitotic cells with dynein deficiencies, and this is not the main point of our current paper. The focus of our current paper is the discovery of mutually exclusive usage of the two light chain paralogs, DYNLRB1 and DYNLRB2, in mitotic and meiotic cells and the contribution of NuMA in their roles. Thus, we think that providing live imaging to further deepen the mechanism of this already reported dynein-deficient phenotype is beyond the scope of our current paper.

Instead, we have cited the preceding paper and added the below sentence to make the additional interpretation clearer.

Results section

“The presence of supernumerary centrioles was reported in DYNC1L1/2-deficient cells as well⁵⁰, and these defects can also be attributed to cytokinesis errors in the previous cell cycle (Fig. 6f)”

5- Figure 6 should be better organized. Currently panel i is presented after j and k.

We have reorganized the figure arrangement.

6- The second to last line is unlabeled in fig 5i.

We have fixed this.

7- A supplementary movie should be provided for the dynein mobility assay presented in fig 6l.

We have prepared Supplementary movie 1 and 2 for GFP-DYNLRB1 and GFP-DYNLRB2, respectively.

8- Line 324: edit “mislocalization”.

We have fixed this.

REVIEWERS' COMMENTS

Reviewer #1 (Remarks to the Author):

The additional clarifications and analyses in the revised manuscript fully addressed this reviewer's concerns.

Reviewer #2 (Remarks to the Author):

The Authors have addressed my concerns adequately.

Reviewer #3 (Remarks to the Author):

The authors have addressed all my concerns in this revised version of their manuscript. It is therefore suitable for publication.